# Carbon footprint distributions of lithium-ion batteries and their materials

Leopold Peiseler [1,2,3] ✉, Vanessa Schenker[4], Karin Schatzmann[1,3], Stephan Pfister [3,4], Vanessa Wood [2,3] & Tobias Schmidt [1,3]

Lithium-ion batteries are pivotal in climate change mitigation. While their own carbon footprint raises concerns, existing studies are scattered, hard to compare and largely overlook the relevance of battery materials. Here, we go beyond traditional carbon footprint analysis and develop a cost-based approach, estimating emission curves for battery materials lithium, nickel and cobalt, based on mining cost data. Combining the emission curves with regionalised battery production announcements, we present carbon footprint distributions (5th, 50th, and 95th percentiles) for lithium-ion batteries with nickel-manganese-cobalt (NMC811, 8-1-1 ratio; 59, 74 and 115 $kg_{CO2}$ $kWh^{-1}$) and lithium-iron-phosphate (LFP; 54, 62, 69 $kg_{CO2}$ $kWh^{-1}$) cathodes. Our findings reveal the dominating impact of material sourcing over production location, with nickel and lithium identified as major contributors to the carbon footprint and its variance. This research moves the field forward by offering a nuanced understanding of battery carbon footprints, aiding in the design of decarbonisation policies and strategies.

Lithium-ion batteries (LIBs) are a key climate change mitigation technology, given their role in electrifying the transport sector and enabling the deep integration of renewables[1]. The climate benefits of LIB-enabled products are evident[2,3], but the production of battery materials[4–7] and the subsequent LIB cell manufacturing[8–10] contribute considerably to greenhouse gas (GHG) emissions—a problem recognised by stakeholders across the battery ecosystem[11–14]. Consequently, policymakers have begun to implement measures to reduce the carbon footprint (CF) of LIBs[14]. These policies are often connected to "green" industrial policy targets of localising battery materials sourcing and cell manufacturing[15,16]. For instance, in 2023, the hallmark European Union (EU) Battery Regulation came into force, coupling market access for EU and non-EU battery producers to battery CFs starting in 2028[11]. Accordingly, leading battery producers recently announced sharp CF reduction targets[12,17]. Furthermore, the US Inflation Reduction Act (IRA) incentivises the localisation of battery material and cell manufacturing[16,18], which in turn affects batteries' CF. Given these policy developments, a

detailed understanding of the drivers and uncertainties of the CF of LIBs is warranted.

There are a multitude of LIB life-cycle assessment (LCA) studies focusing on CF[19–21], but they are known to be sensitive to modelling assumptions and system boundary choices[19,22,23]. Thus, the comparability of LCA studies is still markedly restricted, and interpreting them typically requires domain knowledge. In addition, there are three major knowledge gaps that limit our current ability to comprehend (unintended) consequences and inform the specific design of pioneering policies, such as the EU Battery Regulation, let alone evaluate and benchmark forthcoming decarbonisation initiatives within the battery sector.

First, while the CF impact on energy consumption during battery cell manufacturing is relatively well-researched, only a small number of studies have evaluated the CF of individual battery materials. Furthermore, most of these studies have either analysed individual mining sites or presented commodity-aggregated CF estimates, thus failing to provide a detailed commodity-level picture that includes variance

[1]Energy and Technology Policy Group, ETH Zurich; Clausiusstrasse 37, CH-8092 Zurich, Switzerland. [2]Materials and Device Engineering Group, ETH Zurich; Gloriastrasse 35, CH-8092 Zurich, Switzerland. [3]Institute of Science, Technology and Policy, ETH Zurich; Universitätstrasse 41, CH-8092 Zurich, Switzerland. [4]Chair of Ecological Systems Design, ETH Zurich; Laura-Hezner-Weg 7, CH-8093 Zurich, Switzerland. ✉e-mail: pleopold@ethz.ch

among mining sites (see Supplementary Note 4 for an extensive literature review). Second, battery material CF distributions are not systematically integrated into battery CF analyses. While initial research has begun to aggregate the CF data points of individual battery materials into a battery-level analysis[6,24], the materials' impact and induced variance on battery CF have not yet been quantified. Third, while some studies have analysed the impact of individual battery cell production locations, there is no representative global CF distribution for LIB with nickel-based cathodes (NMC) and iron-based cathodes (LFP). In general, the strong literature focus on NMC is problematic, considering LFP market share projections of over 35% by 2030[25,26].

In this work, to close these knowledge gaps, we develop an approach to quantify material CFs based on cost data. Applying this approach yields emission curves for key battery materials, i.e., mine-level CF estimations and production volumes for lithium, nickel, and cobalt, which we also contrast with findings from a comprehensive literature review. Our insights offer seminal coverage and go beyond existing mine-specific or averaged material CF studies. Furthermore, we use these emission curves and combine them with globally announced battery production locations until 2030 to derive−based on Monte Carlo simulations−the most representative CF distributions for NMC811 (nickel-manganese-cobalt at an 8-1-1 ratio) and LFP LIBs available to date. Lastly, informed by 17 expert interviews, we triangulate and discuss our findings before deriving implications for policymakers and the battery ecosystem.

## Results

### CF of lithium, cobalt and nickel battery materials

The emission curves presented in Fig. 1a, d, g were based on mine-level cost data from S&P Global[27], where our approach translates costs into emissions. Specifically, depending on the mine location and processing steps, we derived unique conversion factors, called dollar emission intensities, that we used to approximate emissions for different cost categories. Unless dotted, every bar in Fig. 1a, d, g represents one mine. Figure 1b, e, h contrast these curves with values from a structured review of the literature and databases. Figure 1c, f, i present global mining production for 2022, categorised by production country, deposit type and data availability and are based on data from the U.S. Geological Survey (USGS)[28], British Geological Survey (BGS)[29], and Deutsche Rohstoffagentur (DERA)[30]. See the Methods section for detailed information.

The emission curve for lithium carbonate depicted in Fig. 1a reveals two primary plateaus: the first, characterised by low CF levels, is predominantly sourced from South American brine operations, and the second, approximately three times higher in CF, is mainly composed of Australian hard rock (spodumene) deposits. Moreover, our analysis showed that while the CFs for brine sites exhibit a wide range, spanning from the lowest to the highest values, all spodumene sites−identifiable by grey bar segments in Fig. 1a−fall within the higher CF plateau.

A large share of the covered supply chain (x axis in Fig. 1a) comprises only a handful of mines with high output volumes. Because many values in the literature and databases refer to South American brine operations, the median literature and database CF is skewed accordingly towards those mines, as shown by the horizontal lines in Fig. 1b. However, looking at the global production locations in 2022, it becomes evident that the actual median CF at around 350 kt of lithium carbonate is about three times higher.

Nickel's emission curve, as depicted in Fig. 1d, can be segmented into three plateaus. The leftmost plateau, accounting for approximately 40% of the covered supply chain, was analysed using the same methodology applied to lithium in Fig. 1a. All mines within this plateau were extracting sulfide ore and corresponded to the non-hatched waterfall segments in Fig. 1f. The CFs in this segment exhibited a hockey-stick curve pattern, with Russia holding the largest market

share at the lower CF levels. For a detailed examination of this first plateau, focusing solely on sulfide mines, refer to Supplementary Fig. 4.

As data for nickel laterite ore mining were not available from S&P Global, it was imputed (dotted in Fig. 1d, f) based on literature, recent reports, and the USGS[28] and BGS[29] datasets. Importantly, for imputed data, individual bars corresponded to country and not to mine levels; see the Methods section for more information. The centre and rightmost plateaus corresponded to two different laterite ore types: limonite and saprolite. They comprised around 54% and 6%, respectively, of the covered supply chain (Fig. 1d's x axis). The CF levels of the second and third plateaus were, respectively, 2 and 7 times the average of the left plateau, although a small number of sulfide deposit mines exhibited higher CF than limonite deposit mines (bars between the centre and rightmost plateaus). The two observed plateaus corresponded to different deposit types, which, in turn, entailed fundamentally different process flowsheets. Similar to lithium, these three plateaus are manifested as a tri-modal probability distribution in Supplementary Fig. 2.

The existing literature and database values were generally in line with the modelled CF of sulfide nickel mining (cf. Supplementary Fig. 4). However, the academic literature and databases do not sufficiently cover nickel from laterite sources and thus underestimate the average CF of nickel sulfate. This gap is likely due to the recent exponential growth in LIB demand, which has resulted in the sourcing of mined laterite for LIBs−an ore deposit that used to be mined predominantly for the steel industry[31,32]. By 2022, Indonesia (home to laterite ore deposits) established itself as the largest nickel mining country, followed by Russia, Canada and Chile (all home to sulfide ore deposits).

The emission curve for cobalt shown in Fig. 1g follows a step-like function, where the four largest mines are all located in the Democratic Republic of the Congo (DRC) and comprise 90% of the covered supply chain. Sites in the DRC mine copper in exclusively stratiform sediment-hosted (SSH) deposits and produce cobalt as a sellable by-product[33]. Cobalt from SSH deposits accounts for the overwhelming majority of the supply chain (Fig. 1i) while by-product cobalt from laterite and sulfide nickel ore deposits plays a minor role.

Similar to nickel laterite, S&P Global data were not available for graphite. Acknowledging Graphite's relevance[34], we used CF ranges from the academic literature and production volumes from USGS[28] and DERA[30]. Panel types a−c are shown for graphite in Supplementary Fig. 3. Graphite's supply chain is dominated by China, where synthetic graphite production accounts for around two-thirds, with slightly higher CF and higher CF ranges than natural graphite. Similar to all previous cases, median database values were lower than median literature values.

### Monte Carlo simulations for LIB with NMC and LFP cathodes

In this section, we present the global representative CF distributions for LIB with NMC and LFP cathodes. Given the opacity of the supply chain, we assumed no regional material preferences among battery producers. For example, we assumed that Chinese and German battery producers were equally likely to source Chilean lithium carbonate. Next, we combined the four emission curves with regionalised announced LIB production capacities until 2030 to obtain CF distributions for LIB cells using Monte Carlo simulations.

For NMC811 cells, whose cathode metal shares comprise 80% nickel, 10% manganese and 10% cobalt, the global CF distribution was bi-modal and characterised by a wide 90% confidence interval, with global CF values ranging from 59 to 115 $kg_{CO_2e}$ $kWh^{-1}$ (Fig. 2a). The left mode was composed of all emission curves, while the second mode was driven by the rightmost nickel sulfate plateau (Fig. 1d). The median CF values ranged from 69 to 77 $kg_{CO_2e}$ $kWh^{-1}$ for Europe and China, respectively (Supplementary Table 3). Importantly, however, the

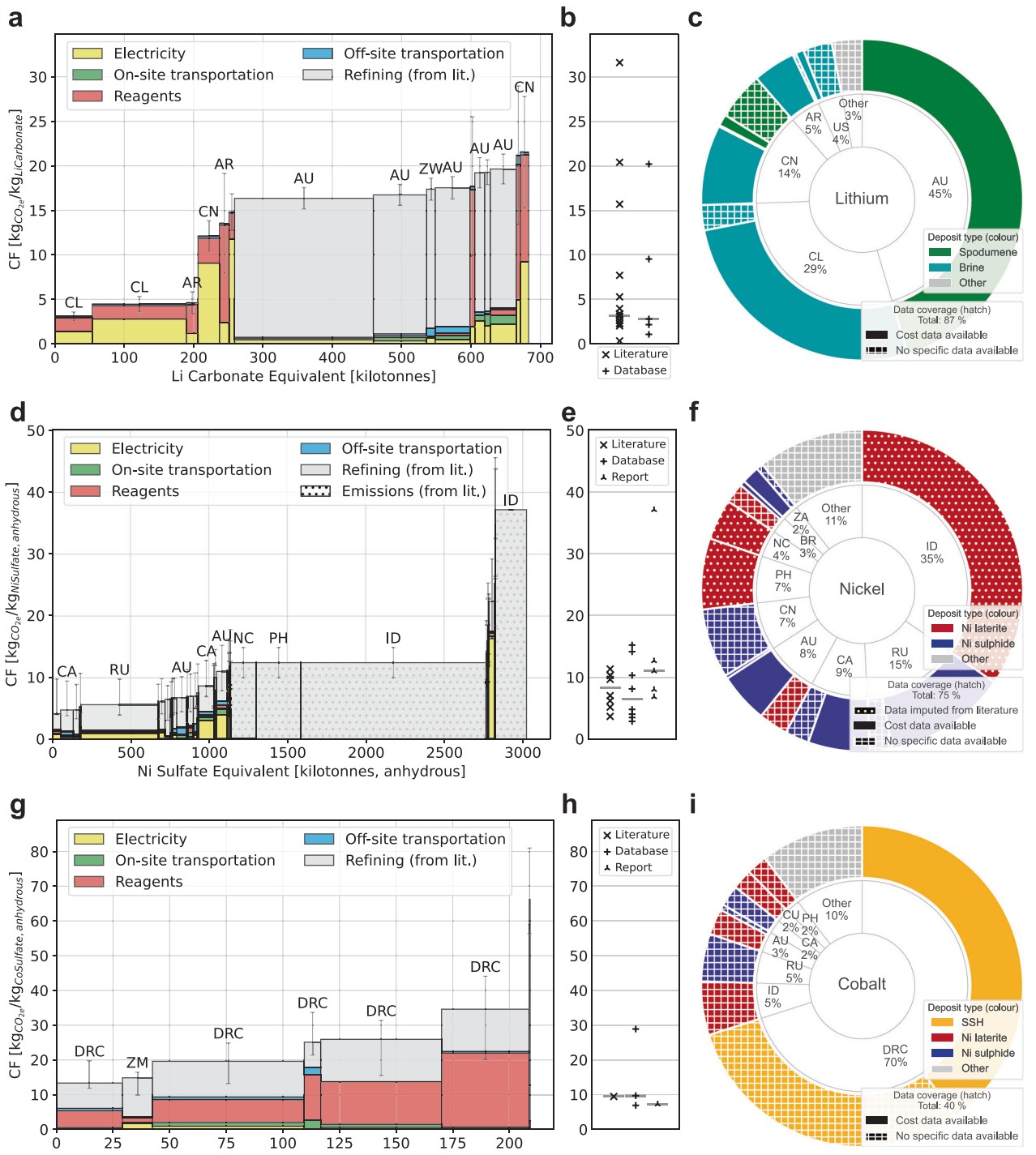

**Fig. 1 | Emission curves, literature and database carbon footprint values and 2022 supply chain for lithium carbonate, nickel sulfate and cobalt sulfate equivalent. a**, **d**, **g** Emission curves for lithium, nickel, and cobalt. Coloured bar segments correspond to modelled data sources, solid grey to literature-based data sources, and dotted grey to imputation-based data sources. Production volumes on the *x* axis are indicated in chemical equivalents, i.e., encompassing all intermediate products that could be refined to battery-grade chemicals. All sulfate products are in their anhydrous form. Whiskers indicate the minimum and maximum values. **b**, **e**, **h** Literature, report and database (ecoinvent 3.9.1 and GREET 2021) values. Horizontal lines indicate respective medians. See Supplementary Note 4 for the underlying review work. **c**, **f**, **i** Doughnut charts of global 2022 battery-specific supply curves, i.e., encompassing all intermediate products that could be refined to battery-grade chemicals. Inner rings refer to country mining shares, while outer rings indicate deposit types and data coverage. Dotted bar segments refer to imputed data, and chequered segments indicate no data availability. Country acronyms: AR Argentina, AU Australia, BR Brazil, CA Canada, CL Chile, CN China, DRC Democratic Republic of the Congo, ID Indonesia, NC New Caledonia (sui generis collectivity of France), PH Philippines, RU Russia, US United States of America, ZA South Africa, ZW Zimbabwe. Other acronyms: SSH Stratiform sediment-hosted Cu-Co deposits. Source data are provided as a Source Data file.

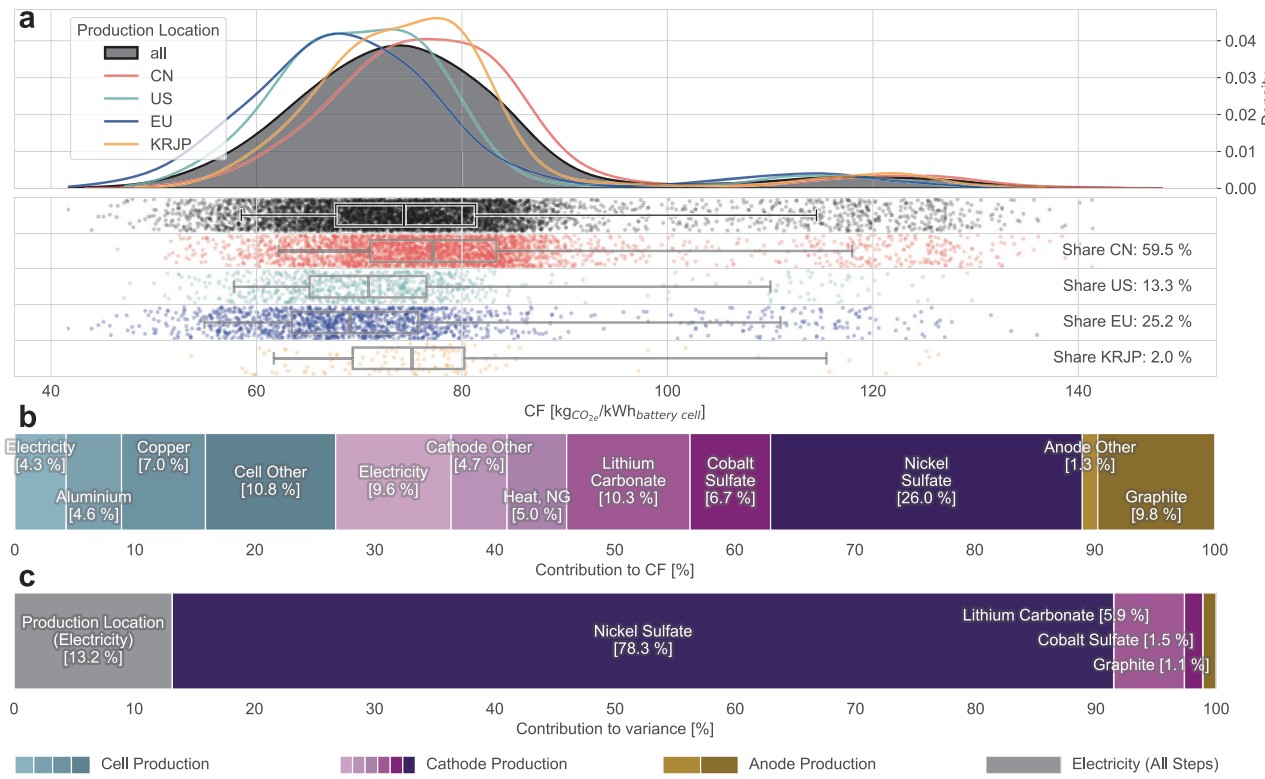

**Fig. 2 | Carbon footprint profile of battery cells with NMC811 cathodes. a** Top: probability density functions for global (black) and regionalised (coloured) production regions. Bottom: Jitter plot with boxplots (90% confidence interval, 5th, 25th, 50th, 75th and 95th percentiles). One dot in the jitter plot represents a conceptual 1 GWh manufacturing plant with a unique combination of production location and CFs of lithium carbonate, nickel sulfate, cobalt sulfate and graphite. See Supplementary Table 3 for the tabulated values. Region acronyms: CN: China, US: United States, EU: Europe, KRJP: Korea + Japan. **b** Relative CF contributions averaged for 100 data points around the global median CF. See Supplementary Table 5 for the tabulated values. **c** Relative variance contributions of sampled parameters for the global Monte Carlo dataset. See Supplementary Table 6 for the tabulated values. **a–c** without nickel laterite mining are available in Supplementary Fig. 7. **a** is also available in a per-kg unit in Supplementary Fig. 5. Source data are provided as a Source Data file.

density functions and dot distributions in Fig. 1a were similar across regions and revealed the subordinate role of battery production location for GHG emissions vis-à-vis material procurement choices. This finding still held if nickel laterite mining was excluded (Supplementary Fig. 7). Moreover, contrary to the widespread perception of high CFs of Chinese LIBs, NMC811 cells produced in China spanned the entire CF spectrum due to announced production in provinces with low-carbon electricity grids.

When we dissected the CF determinants around the global median (Fig. 2b), it became apparent that nickel sulfate played a dominant role. Collectively, active materials made up over 50% of the CF, while contributions from electricity consumption during cell production accounted for around 15% (see Supplementary Table 4 for a detailed breakdown). Adding the contributions of non-active metals, such as copper (7%) or aluminium (5%), highlighted the role of materials even more. To understand CF uncertainty drivers, the global Monte Carlo simulation results were broken down into variance contributions. Figure 2c shows that the CF variance was predominantly driven by nickel sulfate, whose tri-modal probability density function (Supplementary Fig. 2c) is characterised by the three plateaus in its emission curve (Fig. 1d). Results from the analysis without laterite mining (Supplementary Fig. 7) identified production location, nickel sulfate and lithium carbonate as the main drivers, whereas cobalt sulfate and graphite played negligible roles.

For LFP cells, whose cathodes mainly comprised lithium, iron and phosphate, the global CF distribution closely traced the Chinese distribution, as almost all LFP cell production is announced in China (Fig. 3a). The global 90% confidence interval for LFP spanned 54 to

69 kg$_{CO_2e}$ kWh$^{-1}$ (Fig. 3a) and, given the nickel-free cathode, was smaller and more symmetrical than in Fig. 2a. The median CF ranges, at 58–62 kg$_{CO_2e}$ kWh$^{-1}$ for Europe and China, respectively (Supplementary Table 3), were ~16% lower than for NMC cells, including laterite mining. Differences with respect to production locations were even less pronounced compared to cells with NMC cathodes, because less electricity is required for material synthesis and cell production.

Concerning CF distribution, lithium carbonate and graphite were the largest contributors; non-battery-specific materials gained relative shares compared to LIBs with NMC cathodes. The contribution of electricity and, thus, production location noticeably depended on the production method of the active material. Based on expert interviews, we show results for a ratio of 9:1 (solid-state vs. hydrothermal LFP active material synthesis) in Fig. 3 but also provide separate panels a–c for the two isolated synthesis methods in Supplementary Figs. 8 and 9, respectively. Finally, regardless of the synthesis route, lithium carbonate contributed the most, by far, to the observed variance (Fig. 3c).

## Discussion
In this section, we first summarise this work's contribution and contextualise our findings. We continue with an in-depth discussion of the implications for environmental and industrial policy before concluding with an outlook.

This study developed bottom-up modelled emission curves for the battery materials lithium carbonate, nickel sulfate and cobalt sulfate. Due to the different characteristics of each battery material, the shapes (Fig. 1a, d, g) and data coverage (Fig. 1c, f, i) of their emission curves vary noticeably, making broad generalisations inappropriate.

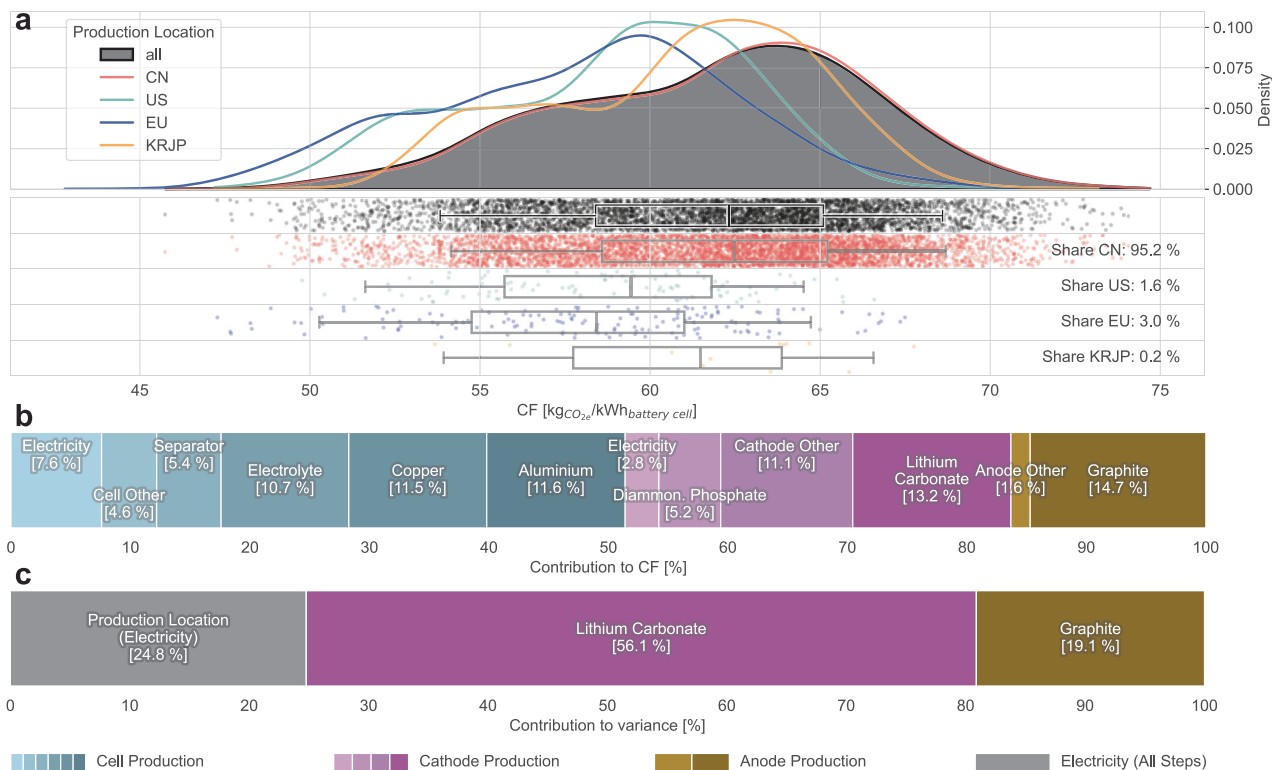

**Fig. 3 | Carbon footprint profile of battery cells with LFP cathodes. a** Top: Probability density functions for global (black) and regionalised (coloured) production regions. Bottom: Jitter plot with boxplots (90% confidence interval, 5th, 25th, 50th, 75th and 95th percentiles). One dot in the jitter plot represents a conceptual 1 GWh manufacturing plant with a unique combination of production location and CFs of lithium carbonate and graphite. See Supplementary Table 3 for the tabulated values. Region acronyms: CN: China, US: United States, EU: Europe,

KRJP: Korea + Japan. **b** Relative CF contributions averaged for 100 data points around the global median CF. See Supplementary Table 5 for the tabulated values. **c** Relative variance contributions of sampled parameters for the global Monte Carlo dataset. See Supplementary Table 6 for the tabulated values. **a–c** are also displayed for individual solid-state (Supplementary Fig. 8) and hydrothermal (Supplementary Fig. 9) LFP synthesis routes. **a** is also available in a per-kg unit in Supplementary Fig. 6. Source data are provided as a Source Data file.

By incorporating these curves into battery cell-level CF analyses and mapping projected battery production capacities, our research provides the most representative CF distributions for NMC811 and LFP LIBs available to date. Our findings highlight the elevated influence of battery materials vis-à-vis cell production locations on CF contributions and variance in distributions. In particular, nickel and lithium emerged as the materials with the largest impacts.

The findings indicate that LIBs with NMC cathodes exhibit higher CF characteristics than their LFP counterparts, particularly when nickel laterite mining is considered. However, this direct comparison should be approached with caution, as calculations were first made on a mass basis before being converted to a capacity basis, as shown in Figs. 2 and 3. Although this study also provides CFs on a per-kilogram basis in Supplementary Figs. 5 and 6 and employs up-to-date cell energy densities for conversion[35], the varying rates and directions of innovation at the cell, module and pack levels for NMC and LFP technologies will affect future CF comparisons.

To put the reported CF distributions into a practical perspective, we built on Sacchi et al.[3] to calculate break-even mileages, i.e., an EV's total kilometres driven at which its lifetime GHG emissions equal those of a comparable combustion engine vehicle. For illustrative purposes, we selected mid-sized SUV electric vehicles (EVs) with built-in NMC batteries from the global 5th and 95th percentiles, respectively. Figure 4a represents the CFs of vehicles when leaving the factory, while Fig. 4b shows lifetime emissions throughout operation. The slopes of the lines correspond to the grid CO2 intensities of the countries in which EVs are operated and the fixed vehicle energy/fuel efficiencies.

In France, a country with low-carbon electricity, the break-even mileages of EVs with built-in 5th and 95th percentile LIBs are around 50,000 km and 80,000 km, respectively (NMC, including laterite mining). Thus, replacing 5th percentile libs with libs from the 95th percentile delays break-even mileage by ~25,000 km. Conversely, in less favourable scenarios, such as in Poland, the break-even mileage is delayed by over 70,000 km, arguably stalling the climate effectiveness of EVs for several years. As this study presumed fixed chemical compositions and manufacturing processes, it captured only a fraction of the real-world variance. Consequently, the true range of CFs and their influence on GHG reduction effectiveness might be even more pronounced. Notably, this simplified calculation underscores the importance of the battery's CF but does not account for the multitude of other parameters[36–38] that influence the real-world lifetime CO2 emissions of thermal and EVs.

The battery and mining industry is approaching a pivotal phase[39], whereby the majority of cell production and mining capacity will be built by 2035[25,40]. While materials from battery recycling are expected to reduce environmental damage, raw material extraction will need to provide the lion's share of battery materials in the foreseeable future[41–47]. As the task ahead involves not merely scaling up mining, refining, and cell production but also ensuring that emission trajectories are directed downwards, a distinct window of opportunity for policy intervention has opened. However, it is crucial that policymakers apply good governance[44] and thoughtfully align environmental and industrial policy to achieve this goal[15].

From an environmental policy perspective, the similarities of regional CF profiles (coloured lines in Figs. 2a and 3a) suggest that

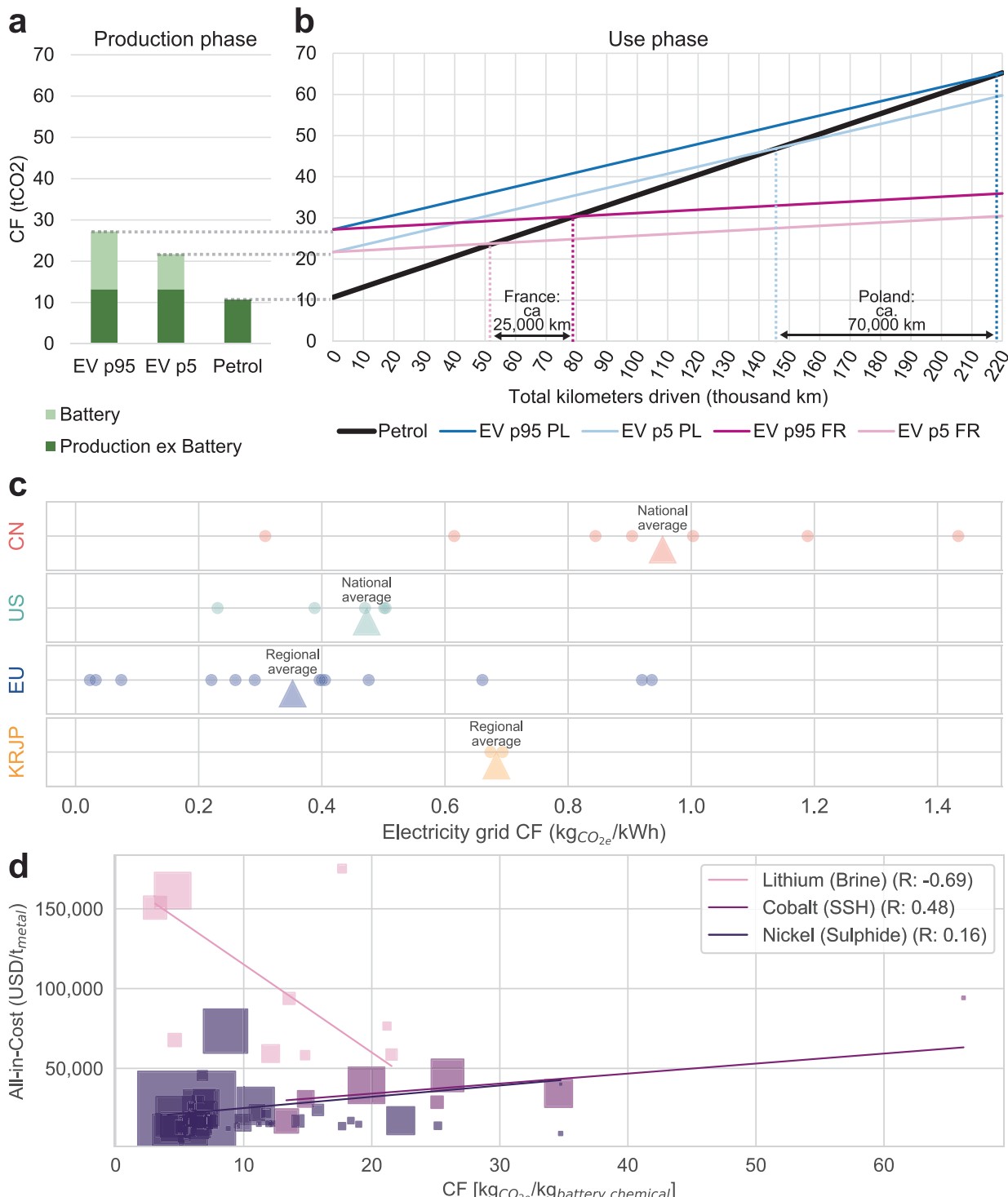

**Fig. 4 | Break-even mileages (total kilometres driven) for electric and conventional vehicles, regional grid CO₂ intensities, and cost vs. emission scatters for lithium, nickel, and cobalt. a** Production $CO_2$ emissions of a petrol vehicle and two electric vehicles (EVs) with built-in batteries from the 5th and 95th CF percentiles of the global CF distribution from Fig. 2a. **b** Life-cycle (production + use phase) $CO_2$ emissions shown for operation in France (pink) and Poland (blue). Break-even mileages are indicated by the dashed vertical lines. For more details, refer to Sacchi et al.[3] and the Methods section. **c** Comparison of grid CO2 intensities of countries and regions for which battery announcements were registered. US states and Chinese provinces were aggregated into regions consistent with ecoinvent's classification. Averages (triangles) refer to: US/China: entire country and not just the regions for which battery plant announcements were registered; Europe: ecoinvent region of Europe without Switzerland; Korean & Japan (KRJP): average of two countries. All values are taken from ecoinvent 3.9.1. **d** Relationship between modelled emissions and all-in costs. In addition to on-site mine costs, which are the basis for the CF calculation, all-in costs also include treatment and refinement charges, royalties, depreciation, and sustaining costs. Trendlines are weighted by mine output volumes, which are indicated by square areas. All-in-cost cost data are only available for lithium from brines, nickel from sulphidic ore, and cobalt from stratiform sediment-hosted Cu-Co deposits (SSH). The index $t_{metal}$ refers to the payable metal content in any refined output product. Source data are provided as a Source Data file.

merely moving manufacturing sites to regions with low-carbon electricity has a limited impact on decarbonisation, as materials constitute the dominant role. Furthermore, product CF policies are scrutinised from a number of angles, including the opacity of supply chains, compliance costs and data confidentiality. This raises an important policy design question: How granular should material emissions data for battery CF calculations be? On the one hand, CFs could be calculated using generic, fixed default values, diluting the metric's significance and missing the opportunity for low-carbon supply chain innovation. On the other hand, requiring detailed, supplier-specific CF data for materials could challenge industry acceptance of policy intervention, straddling the line between precision and practicality.

The proposed EU Battery Regulation[11,48] strikes a balance by providing default values that battery producers can replace with their own calculations, provided they meet specific quality standards. The setting of these default values fundamentally influences the greening effect on the supply chain. If defaults are set at actual median values, half of the material suppliers (i.e., those with higher CFs) will use defaults instead of disclosing their higher CF. To promote transparency and low-carbon innovation, default values should be set at the higher ends of the emission curves. This approach may compromise technical accuracy, but as argued in previous studies[14], there are already existing limitations to the meaningfulness of the CF metric that may justify a focus on promoting low-carbon innovation over strict accuracy. Based on CF contributions and variances, Figs. 2b, c and 3b, c suggest that priorities for addressing knowledge gaps should initially lie in nickel sulfate and lithium carbonate.

From an industrial policy standpoint, Figs. 2a and 3a show that regional differences in the CF profile are not particularly important. With the caveat that battery producers have no regional material preferences, the CF profiles of Chinese-made LIBs largely coincide with those manufactured in the US or Europe. This insight can be explained by the materials' CF (variance) influence and the spatially explicit grid of $CO_2$ intensities beyond national averages. The proposed CF threshold can be visualised as a vertical line in Figs. 2b and 3b, disqualifying all the manufacturing plants (represented as dots) that fall to the right of this line. Unless Western battery manufacturers procure strictly low-carbon materials in large quantities, policy measures such as the proposed CF threshold are unlikely to impact battery industry localisation. Understanding the regional distribution of CFs is essential for assessing the impact of policies both domestically and internationally to avoid unintended consequences, such as cost hikes or damage to the local industry.

The protectionist impact of such policies hinges on the extent to which they mandate the regionalisation of electricity sourcing. If calculations for battery CF are based on national averages (represented by triangles in Fig. 4c) rather than on more localised or sub-regional electricity grid $CO_2$ intensities, manufacturers in Chinese provinces with low-carbon grids (shown on the left in Fig. 4c) are placed at a disadvantage relative to many European producers. In the context of increasingly politically motivated export embargoes on battery materials[49,50], more overt protectionist measures, such as the US Inflation Reduction Act, offer subsidies for batteries and battery chemicals produced locally or in countries with free-trade agreements. No obvious trade-offs between CF and subsidy eligibility emerge when comparing current free-trade agreement countries with the supply chains illustrated in Figs. 1c, f, i. Furthermore, the modest battery production forecasts for Korea and Japan suggest that "friendshoring" may not considerably mitigate the risks associated with battery production.

Considering the global nature of the battery sector, it is necessary to evaluate the global impact of measures such as maximum CF thresholds; the mere reordering of supply chains does not equate with actual decarbonisation progress. As companies start disclosing the CFs of their LIBs, it must be determined whether batteries with lower CFs are more cost-effective than those with higher CFs. If low-CF batteries indeed turned out to be cheaper, they would automatically be favoured by the market, converting green premiums into company profits and rendering any low-CF policy support debatable. Concerning LIB materials, Fig. 4d shows that low-carbon lithium carbonate from brine is more costly than its high-emission counterpart, building a case for policy intervention. While the data are less robust for nickel, the opposite trend is observable for cobalt and nickel, underpinning the complexity of the policy challenge and its (unintended) consequences.

While this work proposes a cost-based methodology and contributes to the breadth and depth of CF analyses of LIBs and their materials, there remains a pressing need for further research. As this methodology does not rely on time-consuming data collection from mining and refining company reports, it can be more easily scaled to additional commodities and panel datasets. Critically, other environmental impacts such as water usage and biodiversity loss of mining through land use change should be quantified with similar granularity[44,51–53]. Applying the proposed cost-based methodology to impacts such as biodiversity is less straightforward than for global warming and thus warrants future work. Additionally, data availability and access continue to pose critical challenges[22,23,54]. Thus, policymakers should ensure that battery CF data, once available, are accessible to researchers in an anonymised but detailed form. For long-term policy planning, it is crucial to consider the multifaceted implications of recycling, ranging from environmental impacts over profitability and technological innovation to supply chain security and geopolitics, requiring future work. Only through continued research can we enhance the understanding of the battery supply chain and its broader environmental effects, setting a precedent for managing the impacts of products beyond global warming.

## Methods
In this study, the term CF was used as a shorthand notation for the 100-year global warming potential (GWP) of a product or process, based on the Intergovernmental Panel on Climate Change (IPCC) 2021 assessment, excluding long-term warming effects[55].

### Material emission curve and supply chain modelling
The battery material selection was based on CF contribution and LIB specificity. Copper, aluminium, phosphate, manganese and iron (sulfate) are not LIB-specific and are used in much larger quantities in other sectors. The production routes for nickel chemicals used in LIBs differ fundamentally from those in the steel industry, the largest nickel consumer[32]. Furthermore, manganese and iron (sulfate) have negligible CF contributions, as shown in Fig. 3b. Therefore, these five materials were excluded from the detailed modelling. Consequently, the materials of interest were lithium, nickel, cobalt and graphite. For the first three materials, bottom-up emission curves were developed, while for graphite and nickel laterite, top-down estimates based on the literature were presented. For methodological considerations regarding the top-down approach, see the "Emission estimations for graphite and nickel laterite" subsection below.

The bottom-up emission curves were based on the 2022 cost data for lithium, nickel and cobalt from the S&P Global Mine Economics Database[27]. The costs were provided on an asset-level in USD per tonne paid metal, converted to USD per tonne metal extracted and broken down into value creation stages (e.g., mine, mill, treatment charges and refining charges (TC/RC), shipment and royalty) and, where applicable, into cost categories (e.g., labour, energy, reagent and other). Because the mining of nickel and cobalt typically produces other economically relevant products, an economic cost allocation ("co-product") was selected. This means that asset costs for common

processes were distributed among the mined metals relative to the metals' final economic revenues.

For all assets (i.e., mine sites), the relevant fields, including geology, output product and processing steps, were manually checked and harmonised by consulting S&P Global's modelling comments, online resources and expert interviews. Consequently, for each commodity, assets were grouped together based on their output product, i.e., the physical product leaving the asset modelled by S&P Global (e.g., concentrated ore or hydroxide). This grouping was necessary to account for different metal contents in the modelled products and thus the varying steps needed to refine them to battery-grade purity. See the Supplementary Methods for detailed information about the cost modelling methodology of S&P Global and property selection.

## Modelling emissions for output products

The fundamental modelling approach of this part of the work was to convert monetary costs into GHG emissions, leveraging asset-specific information, such as location or processing technologies. Because S&P Global cost data covers all steps from extraction to battery-grade chemicals in some cases, but only the extraction steps to intermediate product in other cases, the modelling scope detailed in this section varies accordingly. In cases where cost data did not include steps from intermediate product to battery-grade chemical, our method to account for refining emissions is outlined in the subsequent step (see the "Aggregating S&P Global output products to battery-grade chemicals" subsection).

Practically, the cost categories pertinent to GHG emissions were electricity, on-site transportation, reagents and off-site transportation, while others were less relevant (e.g., labour or royalties). The Supplementary Methods section explains the selected cost categories in greater detail. These cost categories, where applicable, were aggregated over value creation steps (i.e., mining, milling and processing). Deriving conversion factors that we called dollar emission intensity (DEI), the emissions $Em_a$ of asset a was calculated as Eq. (1):

$$Em_a \left[\frac{kg_{CO2}}{kg_{metal}}\right] = \sum_c^C em_{c,a} = \sum_c^C \underbrace{Cost_{c,a} \left[\frac{USD}{kg_{metal}}\right]}_{from S\&P Global} * \underbrace{DEI_{c,a} \left[\frac{kg_{CO2}}{USD}\right]}_{modelled}$$

(1)

where $em_{c,a}$ refers to the emissions arising from cost category c. All cost categories c were contained in C and included electricity, on-site transportation, reagents and off-site transportation costs. Note that $DEI_{c,a}$ was generally asset-specific; that is, it was a function of an asset's locations and processing technologies.

The Excel files in the repository[56] for this paper uniquely identified all mines analysed through S&P Global IDs, providing readers access to the S&P Global database to use these modelling files for reproduction or further analysis. Although this analysis primarily utilised proprietary data for electricity, diesel and reagent prices, as well as grid CO2 intensities, the accompanying repository also contained open-access estimates for these data categories. It is important to note, however, that these open-access estimates were available at a lower spatial resolution and were sourced from various data providers.

## Emissions from electricity costs

We used national average industrial electricity prices (USD MWh⁻¹) provided from Bloomberg New Energy Finance's (BNEF) "Prices, Tariffs & Auctions" interactive datasets for 2022[57]. The CF of 1 kWh electricity from the national grid was taken on a national level from ecoinvent 3.9.1[58]. Consequently, electricity emissions of asset a were calculated as Eq. (2):

$$em_{elec,a} \left[\frac{kg_{CO2}}{kg_{metal}}\right] = Cost_{elec,a} \left[\frac{USD}{kg_{metal}}\right] * DEI_{elec,a} \left[\frac{kg_{CO2}}{USD}\right]$$

$$= Cost_{elec,a} \left[\frac{USD}{kg_{metal}}\right] * \frac{CF_{grid,a} \left[\frac{kg_{CO2}}{kWh}\right]}{nat.elec.price_a \left[\frac{USD}{kWh}\right]}$$

(2)

As we used national figures for prices and electricity CF, note that DEIs for electricity costs were typically only country-specific and not technology-specific. The only exceptions to this were assets that were not connected to the national grid but produced electricity on-site. See the Supplementary Methods for modelling details.

## Emissions from transportation costs

For both on- and off-site transportation, all costs were translated into diesel procurement because this cost category did not include vehicle/infrastructure capital expenditure (CAPEX). Based on Annex 1 of IPCC[59], the CF of diesel combustion was calculated at a global level, as regional differences were negligible for the purpose of this work (see Supplementary Methods). Average national diesel prices (USD l⁻¹) were taken from the same BNEF database as national electricity prices. Analogous to electricity emissions, transportation emissions for asset a were calculated as Eq. (3):

$$em_{transport,a} \left[\frac{kg_{CO2}}{kg_{metal}}\right] = Cost_{transport,a} \left[\frac{USD}{kg_{metal}}\right] * DEI_{transport,a} \left[\frac{kg_{CO2}}{USD}\right]$$

$$= Cost_{transport,a} \left[\frac{USD}{l}\right] * \frac{CF_{diesel} \left[\frac{kg_{CO2}}{l}\right]}{nat.diesel \, price_a \left[\frac{USD}{l}\right]}$$

(3)

## Emissions from reagent costs

While the fundamental principle remained unchanged, determining DEIs for reagents was more complicated than for electricity or transportation. Reagents are chemicals used to process intermediate mining products into battery-grade chemicals. Here, DEIs were determined based on specific reagents used for processing, as well as their prices and CFs. Common reagents were sulfuric acid, calcium oxide (quicklime), or sodium carbonate (soda ash). Typically, processing comprised a flowsheet with multiple steps, each requiring different reagents.

Because S&P Global's reagent costs were not broken down into individual processing steps but were aggregated over all processes, the DEIs also had to capture reagent quantities from all processing steps in a single metric. All assets with the same flowsheet (i.e., processing steps) were assigned a scalar DEI. It was assumed that for assets with the same flowsheet, higher reagent costs did not imply other reagent compositions but larger reagent quantities. Similar to the above, emissions from reagent costs for asset a were calculated as Eq. (4):

$$em_{reagents,a} \left[\frac{kg_{CO2}}{kg_{metal}}\right] = Cost_{reagents,a} \left[\frac{USD}{kg_{metal}}\right] * DEI_{reagents,a} \left[\frac{kg_{CO2}}{USD}\right]$$

(4)

The DEI in Eq. (4), in turn, is asset-specific and was derived from from Eq. (5):

$$DEI_{reagents,a} \left[\frac{kg_{CO_2}}{USD}\right] = \sum_r^R CS_r[1] * \frac{ER_r \left[\frac{kg_{CO2}}{kg_{ore}}\right]}{CR_r \left[\frac{USD}{kg_{ore}}\right]}$$

(5)

where $CS_r$ referred to the cost share, $ER_r$ to the emission ratio and $CR_r$ to the cost ratio of reagent r. Reagent r was an element of set R that comprised all reagents involved in the flowsheet of asset a. The cost

share $CS_r$ of reagent r effectively weighted the term $ER_r/CR_r$ by how much of the reagent costs of one unit dollar overall was spent on reagent r. This was calculated as Eq. (6):

$$CS_r[1] = \frac{m_r \left[\frac{kg_{reagent}}{kg_{ore}}\right] * p_r \left[\frac{USD}{kg_{reagent}}\right]}{\sum_i^R m_i \left[\frac{kg_{reagent}}{kg_{ore}}\right] * p_i \left[\frac{USD}{kg_{reagent}}\right]} \qquad (6)$$

where $m_r$ referred to the quantity of reagent r needed for processing one kg of ore and $p_r$ referred to the unit price of reagent r. For every processing step in a flowsheet, $m_r$ was determined by consulting metallurgic literature and books; where necessary, it was corroborated by lab protocols and expert assessments. The price $p_r$ of reagent r was estimated by internet research, with a preference for 2022 bulk volume figures. Both parameters were treated as global, but regional and temporal variations were considered in the Material Monte Carlo simulation (see the "Uncertainty quantification–material Monte Carlo simulation" subsection).

The emission ratio $ER_r$ was determined by Eq. (7):

$$ER_r \left[\frac{kg_{CO2}}{kg_{ore}}\right] = m_r \left[\frac{kg_{reagent}}{kg_{ore}}\right] * CF_r \left[\frac{kg_{CO2}}{kg_{reagent}}\right] \qquad (7)$$

where $CF_r$ refers to the CF of reagent r (i.e., the GWP per unit mass of reagent r). To determine this figure, we matched reagent r with the ecoinvent 3.9.1. database and, where necessary, determined chemically similar proxies. As usual, the IPCC 100-year GWP impact category was used to determine the reagents' CFs from ecoinvent.

Analogous to Formula (7), the cost ration $CR_r$ for reagent r was calculated as Eq. (8):

$$CR_r \left[\frac{USD}{kg_{ore}}\right] = m_r \left[\frac{kg_{reagent}}{kg_{ore}}\right] * p_r \left[\frac{USD}{kg_{reagent}}\right] \qquad (8)$$

This entire section was operationalised in Microsoft Excel, where emissions for one commodity (lithium, nickel and cobalt) were modelled in separate files. See the Supplementary Methods for details on this step of the emission modelling.

## Uncertainty quantification–material Monte Carlo simulation

To quantify the modelling uncertainty of the above CF calculations, we used the Excel plugin @risk. This tool allows input parameters to be represented by distributions instead of scalars and propagates the statistical variation through Excel formulas to the result cells. For all input parameters, we defined uncertainty distributions based on expert interviews, our own simulations or multiple data sources. Where only single data points were available, we assumed triangular distributions with ±5% (diesel prices) and ±10% (reagent prices). See Supplementary Table 1 for a breakdown of the input distribution assumptions and for further details. Finally, for $n = 5000$, the tool simulated the output products' emission distributions for all mining assets.

The 5th, 50th and 95th emission percentiles of the MC simulation were propagated to the Python environment that was used for further analyses. For every asset, the difference between the 95th and the 5th percentile defined the margin of error regarding the modelled output product, while the 50th percentile was used as the most likely value.

## Aggregating S&P Global output products to battery-grade chemicals

The emissions of all output products that were not yet battery-grade chemicals had to be complemented to allow for a fair comparison between assets. To this end, an extensive structured literature review of lithium, nickel, cobalt and graphite was conducted. The backbone of the review was built on Scopus, the Google search engine and connectedpapers.com. Furthermore, the output of the initial search was expanded through forward and backward references based on all previously identified sources. See Supplementary Note 4 for the detailed methodology and results of the literature review.

Having identified all pertinent work related to battery material emissions, in the next step, we disaggregated the reported CF estimates into different value creation steps wherever the original study allowed for this. The emission categories used were Mining and Beneficiation, Transport, Refining and Other. For each commodity and output product, a separate Excel spreadsheet listed the disaggregated emissions for all eligible publications. Ultimately, this step yielded an emission range that could be used to quantify additional emissions needed to refine the output product to battery-grade chemicals.

As shown by the grey bars in Fig. 1, refining emissions constituted a large share of material emissions. Since the findings from the literature review were not asset-specific but rather commodity-specific, we leveraged TC/RC to improve our best estimate for asset-specific refining emissions. TC/RC were provided for all assets by S&P Global and can be understood as the asset-specific cost of turning the output product into a high-purity product (e.g., refined metal). Typically, the higher the concentration of undesirable elements, the higher the required process engineering and energy efforts and thus the higher the TC/RC. See the Supplementary Methods for further information on TC/RC.

The refining emission range was determined by the minimum and maximum values of the disaggregated literature emission spreadsheet and was identical for all assets of the same product group and commodity. Since TC/RC have some explanatory power about refining energy and thus emissions, we based an asset's most likely refining emissions on its relative TC/RC cost. Specifically, we coupled assets' TC/RC range (minimum and maximum) with the refining emission range in a linear fashion and estimated an individual asset's most likely refining emission based on its TC/RC position on the TC/RC range. This meant that the asset with the lowest TC/RC was assigned the lowest most likely emissions from the emission range, vice versa and everything in between. Given that TC/RC was not a perfect proxy for emissions, the reserved lower (upper) error for refining emissions of all assets was the difference between the most likely emission estimate and the lowest (highest) value of the emission range. Consequently, all assets had the same minimum and maximum refining emissions and only the most likely value was determined by their TC/RC. By design, this methodology led to asymmetric margins of error, which can be seen in Fig. 1. The resulting refining emissions and refining margins of error were added to the output product emissions and margins of error. For lithium, S&P Global did not provide TC/RC, which is why—for all assets—the minimum and maximum literature values were used for the margin of error and the literature average for the most likely value.

As the ecoinvent battery cell inventory built on Dai et al.[60], who, in turn, reported inventory data for anhydrous sulfate forms, in this work, we also reported the quantities and CFs of sulfate compounds with respect to their anhydrous form. Literature values that reported sulfate compounds in hexa- or heptahydrate forms were stoichiometrically converted to anhydrous sulfate compounds.

## Estimating emissions and production volumes for graphite and nickel laterite

S&P Global does not provide data for graphite and no usable asset-level data for nickel laterite mining. However, as both materials are known to drive the CF of batteries[5,24,34], it was decided to include their emission ranges in the battery CF Monte Carlo simulation based on country-specific secondary literature. For graphite, we utilised three publications reporting CFs of both natural and synthetic graphite (Supplementary Note 4). Nickel laterite, on the other hand, was further divided into limonite and saprolite mining, whose CF estimation built—given the lack of peer-reviewed work—on one database value (GREET)

and two consulting reports that were both carried out by LCA consultancy Minviro using their proprietary inventory (Supplementary Note 4). It should be noted that peer reviews for one of the reports[61] were conducted by industry experts and are accessible at the end of the document. Analogous to Fig. 1, the emission curves, including literature values, and the supply chain mapping for graphite and nickel, including laterite, are shown in Supplementary Figs. 3 and 4, respectively. National production volumes were taken from the battery-specific value chain mapping described in the method section "Constructing and mapping global battery supply chains" and in the Supplementary Methods section.

## Comparing modelled emissions to literature and database values

An extensive structured literature review formed the basis for Fig. 1b, e, h. If one study reported multiple material CFs based on, for example, multiple production locations or processing routes, all estimates aligned with the selection criteria were transferred into our database. We labelled studies that were pertinent but, for example, lacked transparency or represented poor current or near-future battery material production as "peripheral" and excluded them from our analysis. Next to a detailed description of the literature review methodology, Supplementary Note 4 provides the database with qualified and peripheral studies alongside case-by-case exclusion criteria. Beyond peer-reviewed and other academic documents, database values were collated from GREET 2022[62] and ecoinvent 3.9.1[58] for all available pathways.

## Constructing and mapping global battery supply chains

The bottom-up modelling efforts described above and shown in Fig. 1a, d, g are contrasted with top-down national mining production shown in Fig. 1c, f, i to better understand the coverage and limitations of S&P Global data. To this end, reported production data were retrieved from USGS[28] for 2022 and supplemented by BGS[29] and DERA[30] where necessary. To build a global battery-specific commodity supply chain, we considered all mining (intermediate) products that could potentially be used in batteries. In other words, the x-axis in Fig. 1a, d, g does not represent the actual volume of battery chemicals but rather the volume from which battery chemicals could be produced and supplied to battery manufacturers. This is denoted by the equivalent in Fig. 1's x-axis labels. This differentiation is relevant for nickel, as for some fractions of mined nickel, there is a commercial pathway for producing nickel sulfate. See the Supplementary Methods for further information and calculations.

## Battery CF distribution

The selection of cathode chemistries for this work (NMC811 and LFP) was motivated by market forecasts that attribute high-nickel and iron-based cathodes to the most dominant market shares[25]. Close relatives to the selected cathodes, e.g., NMC955 (90% nickel and 5% each of manganese and cobalt) or lithium manganese iron phosphate, were not explicitly modelled, as small changes in the bill of materials would only slightly change this work's findings.

## Converting emission curves to probability distributions

In preparation for the battery Monte Carlo simulation, the emission curves from Fig. 1 were converted to probability density functions (PDF), which, in turn, were used for parameter sampling during the Monte Carlo simulation. Conceptually, this step assigned the CF of every asset in the emission curve a likelihood based on the asset's production volume. Practically, for every commodity, random (but weighted by the assets' production volumes) samples ($n = 10,000$) were drawn from the emission curve and used for non-parametric Gaussian kernel density estimation. Incorporating the uncertainty in the emission curves, a triangular distribution was selected with the

most likely asset emission as the mode and the asymmetrical margins of error as minimum/maximum. The Supplementary Methods contains the PDFs for the four commodities, including nickel with and without laterite production.

## Collating regional announced battery production capacity

We built our battery production database using an interactive BNEF dataset[57] and including operational LIB production sites and those announced to be operational by 2030. The raw data were classified by nameplate capacity, produced battery chemistry and country. In the first step, the data were filtered for NMC, LFP or unknown battery chemistries. Since BNEF discloses production sites only at the country-level, we turned to the Supplementary Information of Kallitsis et al.[6], who mapped the announced LIB production at the provincial (China) and state (US) levels through 2035. From there, the provincial/state shares were taken and applied to the total announced production capacity of China/US provided by BNEF. In the next step, CN provinces and US states were grouped and harmonised with ecoinvent 3.9.1 electricity grid regions. For Europe and KR/JP, no regional disaggregation was possible, as these countries occupy the smallest available spatial unit in ecoinvent. In the last step, we imputed the information on regional chemistry prevalence with an expert interview that generally fell in line with the regional NMC:LFP shares derived from BNEF. We ultimately arrived at a provincial- (CN), state- (US) and country-level (Europe, JP and KR) understanding of the announced LIB production capacity of NMC and LFP through 2030. The regional terms European Union and Europe were used interchangeably throughout this document. The United Kingdom and Serbia were the only non-EU countries that were counted towards the announced European battery capacity.

## Setting up battery Monte Carlo simulations

The Python library Brightway2[63] is at the heart of the battery Monte Carlo simulation of this work. Please refer to the Supplementary Methods for a detailed description of how the environment was set up and operationalised. Conceptually, we use the Ecoinvent 3.9.1 bill of material (BOM) for NMC811 and LFP LIB cells and build our own foreground database incorporating all activities from LIB cell production to the point of battery chemistry intake (lithium carbonate, nickel- and cobalt sulfate, and battery-grade graphite). Consequently, in our modelling, the active material synthesis occurs in the same electricity grid region as the battery cell production. The final Brightway2 database file is available in the accompanying repository. The CF of the battery-grade chemicals is defined as parameters and coupled to the probability density functions derived earlier. Further, we also define the location of electricity exchanges as a parameter and couple it to the announced battery capacity database. Lastly, a custom Monte Carlo function was developed that can selectively vary battery chemical and location parameters. Note that there are two important assumptions here: Firstly, we assume a global commodity market where, e.g., Chinese battery producers are equally likely to source lithium carbonate from Chilean mines compared to Australian-mined and Chinese-processed lithium carbonate. Secondly, for our entire work, a constant BOM is used, i.e., all material and energy requirements are kept constant and only the CF of four materials and the electricity is varied. The functional unit of simulation output is kg, which is turned into kWh in the last step. The energy densities for conversion are taken as scalars from recent work[35] and are in line with the original work[22] from 2021 that, in turn, builds the basis for the current Ecoinvent inventory (NMC: 0.25 kWh kg$^{-1}$ and LFP: 0.16 kWh kg$^{-1}$). In Supplementary Note 3, the results of Figs. 2 and 3 are displayed in kg units.

Throughout our analysis, the sample size was set to $n = 11,308$, aligning with the announced overall nameplate battery production capacity by 2030 in GWh. Lines in Fig. 2a and Fig. 3a are probability

density functions using the same approach described above. If necessary, additional region-specific MC runs were executed to ensure a sample size of at least $n = 1000$ for each regional probability density function. The number of dots in the jitter plots (Fig. 2b and Fig. 3b) corresponds to the chemistry's respective announced production 2030 capacity (5797 GWh for NMC and 5,510 GWh for LFP), whereas the boxplot statistics refer to the entire sample size (11,308). To keep computational requirements manageable, the four probability density functions were linearised with a step size of $n = 100$ and used for all Monte Carlo runs presented in this work. Deviations between the computationally intensive and less intensive, linearised-PDF sampling remain at sufficiently low levels at <2%.

The CF disaggregation shown in Figs. 2c and 3c was based on 100 data points around the median. Specifically, for the subset of these MC runs, the commodity parameters were first averaged and then combined with the minimum and maximum CF grid locations, respectively. The breakdown was done once for the minimum and maximum CF grid locations (both using the same averaged commodity parameters) and then linearly combined to yield the median CF.

The variance quantification and disaggregation shown in Figs. 2c and 3c were based on the assumption that each parameter is stochastically independent, i.e., the procurement choice of one commodity has no impact on the choice of another commodity. Using this property, the sum of individual variances of the Monte Carlo simulations with only singular parameter variations was equal to the total variance of the Monte Carlo simulation, where all parameters are sampled simultaneously. This additivity property was used in other work[64] and mathematically backed in the Supplementary Methods.

## GHG break-even point based on CF distribution and EV operation locations

We used the Excel tool available from Sacchi et al.[3] to calculate mileage break-even points for EVs and internal combustion vehicles. Consequently, see the original publication for assumptions, system boundaries and further methodological details. In their Excel tool, we modified the CF of the battery cells according to the 5th and 95th percentiles of the global distribution shown in Fig. 2a. For calculation details, see the Excel file "Breakeven_calculation_BEV_ICE.xlsx" in the repository.

## Expert interviews

Over the course of this work, 17 expert interviews were conducted to iteratively develop and refine the research design, triangulate assumptions, and corroborate the findings and implications of this work. Eligible experts from academia, industry, and policy were identified through publication authorship, conference attendance, social media, and snowballing. Interviews were generally conducted online. They lasted between 30 and 60 minutes and were semi-structured (i.e., a set of guiding questions was used). Interviewees were typically sent the set of questions and preliminary research findings beforehand. Where participants agreed, the interviews were recorded and transcribed for further analysis. See the Supplementary Methods for an anonymised list of the interviews. Since the purpose of the interview process was to elicit feedback on the research design and results, it did not involve personal data. It was confirmed by the Secretary General of the ETH Zurich Ethics Commission that such interviews do not represent research involving the data of human participants.

## Reporting summary

Further information on research design is available in the Nature Portfolio Reporting Summary linked to this article.

## Data availability

The raw data from S&P Global, BNEF, and ecoinvent are protected for proprietary reasons but can be obtained by contacting the corresponding author and with permission of the respective data provider. Open-access data alternatives for BNEF and coinvent are provided in the repository. The data generated in this study are provided in the repository and in the Source Data file. Source data are provided in this paper.

## Code availability

A repository encompassing the computer code, Excel files, and non-proprietary data used in this study is available at Zenodo under https://doi.org/10.5281/zenodo.13936832[56].

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

## Acknowledgements

This article was written as part of the SWISSCHAINS project of the Institute of Science, Technology, and Policy at ETH Zurich. This work was done within the project "e-Bike City". Funding by the Civil, Geomatics, and Environmental Engineering Department of ETH Zürich is gratefully acknowledged. The authors would like to thank all the interviewees for their valuable time and contributions to this study. They also express their gratitude to Julian Rieder for research assistance, Dr. Aleksandra Kim for Brightway2 support, Dr. Romain Sacchi for help regarding life-cycle vehicle emissions and Ketan Vaidya and Achim Teuber for their ongoing feedback. The authors also thank Paul Waidelich, Dr Bessie Noll and other members of the Energy and Technology Policy Group at ETH for very helpful feedback on earlier drafts of the paper. ChatGPT was utilised for assistance with language and Python coding.

## Author contributions

L.P., V.S., S.P., and T.S. conceptualised the work and developed the methodology. K.S. curated the data. L.P. conducted the formal analysis. V.W., S.P. and T.S. supervised the work. V.W., S.P. and T.S. acquired funding. L.P. wrote the original draft.

## Funding

## Competing interests

The authors declare no competing interests.
