## [Peer Review File · Nature Communications]

REVIEWER COMMENTS

Reviewer #1 (Remarks to the Author):

This research provides results on carbon footprints of different salts for lithium ion battery. It is interesting to combine cost into the carbon footprints analysis. The reviewer has the following comments:

1. It is necessary to clearly explain Fig. 1. The results seem not correct referring to the calculation raw data. Why the CF is low for CL while it is high for CN concerning lithium carbonate for instance? The source data are not convincing. They need to be verified.
2. Figure 2 is also not correct. Concerning NCM811 the production rate given in the figure is not precise and the data can vary. For instance, the NCM811 materials can be produced in one country while the battery is produced in another. Where is the electrolyte?
3. Figure 2b the share of electricity has two parts. What are the difference? in Fig. 2c, the share of contribute to variance for NiSO₄ dominates. It is necessary to provide reasons and how much the divation will be.
4. Fig. 3 has the same problem.

Reviewer #2 (Remarks to the Author):

The paper "Carbon Footprint Distributions of Lithium-Ion Batteries and Their Materials" addresses a significant aspect of environmental impact assessment in the context of lithium-ion batteries (LIBs). The study focuses on the carbon footprint (CF) associated with LIBs, particularly the contributions from the sourcing and processing of key materials such as lithium, nickel, and cobalt.

The paper employs a novel cost-based methodology to estimate emission curves for lithium, nickel, and cobalt, which allows for a more nuanced understanding of the CF associated with these materials. This approach adds some value by highlighting the variance in emissions based on different mining sites and processing methods.

The paper is quite interesting. However:

- Some problems rely on proprietary data that cannot be shared, mainly the mining data. This makes it extremely hard to better understand and reproduce the analysis.

- Please reconsider the bibliography: BloombergNEF. Electric Vehicle Outlook 2022, can be substituted with the 2024 version (<https://about.bnef.com/electric-vehicle-outlook/#download>)

- In addition consider the following report: Global EV Outlook 2024 <https://www.iea.org/reports/global-ev-outlook-2024>. This is very interesting because it shows very similar results for LFP batteries. Then the basic data used for the report are affordable. Please comment.

- I'm quite surprised about the NMC811 results, which are lower than those expected. Check Global EV Outlook 2024 <https://www.iea.org/reports/global-ev-outlook-2024> and justify.

Minor comments:

- The paper assumes no regional material preferences among battery producers, which might oversimplify the complexities of global supply chains.

- Although the paper acknowledges the potential of battery recycling to reduce environmental impacts, it does not extensively explore this aspect. Including more detailed scenarios on the role of recycling in CF reduction could provide a more holistic view.

- The focus on CF is crucial, but a more comprehensive environmental impact assessment including factors like water usage, land use change, and biodiversity loss could provide a more complete picture of the environmental footprint of LIBs.

Reviewer #2 (Remarks on code availability):

The results are not reproducible. Several data are not available

Reviewer #3 (Remarks to the Author):

In recent years, several countries around the world have issued policies and regulations to encourage the localization of the battery industry chain. This study develops a novel cost-based approach to estimate emission curves for the key battery materials based on mining cost data. The

research is both interesting and meaningful. However, in my view, there are still some critical issues that need to be addressed.

1. Abstract. The analysis of the background is too long and lacks quantitative results. It is recommended that the author rewrites the Abstract.

2. Introduction. The greenhouse gas emissions on battery production, please consider:

a) 10.1016/j.etrans.2022.100169

b) 10.1016/j.jclepro.2022.133342

3. Figure 1b, e, f is unclear and it is recommended to change the fill color. Additionally, it is suggested to supplement the x-axes in Figures 1c, f, and i.

4. The synthesized graphite produced in China is characterized by high carbon emission profiles, and it constitutes about two-thirds of the global output. Why are the median values from the database lower than the median values from the literature?

5. The synthesized graphite produced in China is characterized by high carbon emission profiles, and it constitutes about two-thirds of the global output. Why are the median database values are lower than median values from the literature?

6. In numerous literature sources, the proportion of active substances is approximately one-third (for example, Ciez and Whitacre, 2019, Nature Sustainability 2(2), 148-156). Why does this study find that active substances account for more than 50% of the carbon footprint?

7. From Figure 2c, it can be observed that the contribution of nickel sulfate is 77.5%. The authors' conclusion that "the CF variance is virtually exclusively driven by nickel sulfate procurement" is overly absolute.

8. Line 203. The reviewer did not find Figure 3d in the manuscript.

9. Figure 3a shows that the most LFP is produced in China. The cathode current collector commonly used in batteries is usually aluminum foil, which is produced in China through an energy-intensive and high-emission process of electrolytic aluminum, often powered by coal-electricity. Figure 3c reveals that the carbon footprint of LFP is primarily contributed by lithium carbonate, electricity, and graphite. However, the absence of a contribution from the high-emission electrolytic aluminum process is puzzling.

10. Line 219-221. Please provide evidence for the statement "the shapes and data coverage of their emission curves vary significantly."

11. Line 234-236. Why did the authors choose mid-sized SUV electric vehicles with built-in NMC batteries? What is the chemical system of NMC batteries?

12. Line 273. What exactly is "the evidence"? Are there any referenced literature or reports? Please list them.

13. The emissions method proposed in the abstract of this paper, which is associated with cost, is barely addressed in the discuss section.

14. Line 294-297. Without considering upstream materials, "the CF profiles of Chinese-made LIBs largely coincide with those manufactured in the US or Europe" seems to differ from current research.

15. Line 323-324. Please confirm whether the "low-carbon lithium carbonate from..." mentioned is indeed represented in Figure 4b. Verify the entire text and make any necessary revisions.

16. Line 344-346. Nickel is also not LIB-specific and used in much larger quantities in stainless steels. Why does the author focus on studying it?

17. Line 452-454. It is suggested that the content of "expert interviews and educated guesses" be supplemented to enhance credibility.

18. Line 567-578. Please provide the data on the provincial- (CN), state- (US), and country-level (Europe, JP, KR) understanding of announced LIB production capacities of NMC and LFP until 2030.

No.	REVIEWER COMMENT	RESPONSE New or amended passages are shown in bold blue . Page and paragraph references correspond to the revised manuscript with tracked changes.
REVIEWER 1		
1.1	This research provides results on carbon footprints of different salts for lithium ion battery. It is interesting to combine cost into the carbon footprints analysis. The reviewer has the following comments:	We thank the reviewer for their time and insightful feedback on our manuscript. We value their comments as they help us enhance the clarity and impact of our research.
1.2	It is necessary to clearly explain Fig. 1. The results seem not correct referring to the calculation raw data. Why the CF is low for CL while it is high for CN concerning lithium carbonate for instance? The source data are not convincing. They need to be verified.	We gladly clarify the methodology for deriving the carbon footprints (CF) shown in Figure 1. Generally, the differences in CFs of mines can be explained by walking through the formulas provided in the Method section of the manuscript. For example, as explained in Formula (1), the CF for each mine is calculated as the sum of individual CF contributions from various cost categories, such as electricity, diesel, or chemical reagents. Each contribution is influenced by the mine-specific cost (e.g., 100 USD spent on diesel per 1 kg of lithium carbonate) and the “Dollar Emission Intensity” (DEI) (e.g., 1 kg of CO₂ per 1 USD spent on diesel). These costs and DEI values are typically specific for each mine and cost category. Additionally, where mine-specific data from S&P Global only covers part of the processing chain (e.g., up to lithium concentrate for spodumene-mined lithium), we have incorporated emissions from refining to battery-grade material using literature values. We refer to the method section and the attached Supplementary Information (SI) for detailed explanations. Regarding the data sources and their verification, we acknowledge the importance of data reproducibility and open access in academia. Unfortunately, the seminal scope of our study (both breadth and depth) necessitated the use of proprietary data that we cannot disclose. To mitigate concerns regarding transparency and reproducibility, we have implemented the following changes:  1. Our source files now include unique identifiers and URLs to all individual S&P Global Mine Economics assets, allowing anyone with access to the S&P Global Capital IQ Platform (available at many academic institutions) to retrieve the underlying cost data. 2. To ensure operability of our method and code for readers without access to the proprietary data bases, our revised repository now features additional open-access data sources for electricity prices for industrial consumers (statista.com and globalpetrolprices.com for BloombergNEF data), diesel prices (iea.org and

		globalpetrolprices.com for BloombergNEF data) and grid CO2 intensities (ourworldindata.com for ecoinvent). While these sources provide insights similar to the proprietary data, they are less spatially detailed, which is why we rely on proprietary data for our analysis. 3. We have also added an Excel calculation file to the repository, substituting proprietary S&P Global data with illustrative data for two generic lithium mines. Using the open-access data sources from above (2), this new file allows the replication of our method for determining mine-level CFs. To deal with input data ranges and uncertainties, the Monte Carlo simulation in Excel requires the @RISK add-on tool, which is available for a free 15-day trial. This Excel file contains all relevant CF contributors at the mine level, but the Python code provided in the repository (02a_Create_Emission_Curves.ipynb) must be executed to add refining emissions (if necessary) and to produce visualisations similar to Figure 1a. In terms of data reliability, our source data from S&P Global is extensively used by industry and has been validated in expert interviews. Furthermore, the CF values determined using our novel methodology align well with conventional mine-specific CF estimates. Additionally, in the uncertainty calculations for the CF of battery materials (Figure 1), we apply a $\pm 10\%$ triangular distribution to all data used from S&P Global (see SI Section 1.5). To address the reviewer’s concern, we added the following paragraph to our manuscript: page 16, line 428 The Excel files in the repository for this paper uniquely identify all mines analysed through S&P Global IDs, enabling readers with access to the S&P Global database to use these modelling files for reproduction or further analysis. Although this analysis primarily utilises proprietary data for electricity, diesel, and reagents prices, as well as grid CO2 intensities, the accompanying repository also contains open-access estimates for these data categories. It is important to note, however, that these open-access estimates are available at a lower spatial resolution and are sourced from various data providers.
1.3.1	Figure 2 is also not correct. Concerning NCM811 the production rate given in the figure is not precise and the data can vary. For instance, the NCM811 materials can be produced in one country while the battery is produced in another.	We thank the reviewer for their attention to the details in Figure 2. The production shares are indeed not “precise” but intentionally represented as probabilistic, thus reflecting the inherent diversity in production locations. These values are derived from a Monte-Carlo simulation involving over 11,000 runs, grounded in proprietary Bloomberg NEF data and corroborated by the findings of Kallitsis et al. (2024), as discussed in the Collating regional announced battery production capacity method section of our manuscript.

		We appreciate the reviewer's concern regarding the geographical discrepancies between the production of NCM811 materials and the assembly of battery cells. This work's research design explicitly accounts for the reviewer's remark: in the Monte-Carlo simulation, we not only vary the location of the cell manufacturing site (paragraph above) but also the provenance of the material. Specifically, we build on Figure 1 to derive probability density functions (SI Figure 2) that are used as underlying material distributions for the Monte Carlo algorithm. This approach ensures a robust analysis of global production dynamics.
1.3.2	Where is the electrolyte?	The carbon footprint contribution of the electrolyte, quantified at 3.24 kgCO₂/kWh, is indeed included in our analysis but aggregated under the "Cell Other [10.8 %]" category in Figure 2b for clarity and visual succinctness (resp. [4.6 %] in Figure 3b). For a detailed breakdown of each component's carbon footprint, including the electrolyte, the reviewer is referred to SI Table 4 where we present the disaggregated data.
1.4.1	Figure 2b the share of electricity has two parts. What are the difference?	We thank the reviewer for highlighting this. The separation of different electricity contributions in Figure 2b aims to illustrate the distinct value chain steps where electricity is consumed. Notably, a significant portion of electricity is required for producing the cathode, especially during the calcination of NMC hydroxide to NMC oxide. In the initial submission, different colour schemes were used to represent cell-level (blue), cathode (lilac), and anode (brown) processes. However, a colour legend was missing in the Figure panel. To address the reviewer's helpful comment, a colour legend has now been added to enhance clarity in revised Figure 2 (and Figure 3 and SI Figures 7, 8 ,9).
1.4.2	in Fig. 2c, the share of contribute to variance for NiSO ₄ dominates. It is necessary to provide reasons and how much the divation will be.	We thank the reviewer for highlighting that the dominant contribution of nickel sulphate to the overall variance in Fig. 2c was not clearly explained in our initial submission. The mechanistic reason for this dominance is that the underlying probability density function for nickel sulphate is tri-modal over a large domain, as shown in Nickel incl. laterite in SI Figure 2 (third column). Since variance is calculated by considering the squared differences between the dataset's mean and its individual data points, a distribution with pronounced outliers will lead to large variance. SI Table 6 contains numerical results for variances shown in the main manuscript. Following the reviewer's recommendation, we have added the following explanation: page 6, line 176 Figure 2c shows that the CF variance is predominantly driven by nickel sulfate, whose tri-modal probability density function (SI Figure 2c) is characterised by the three plateaus in its emission curve (Figure 1d).

1.5	Fig. 3 has the same problem.	We agree with the reviewer's helpful comment. In light of our responses to comments 1.4.1 and 1.4.2, we have made analogous amendments to Figure 3 as well.
-----	------------------------------	---

No.	REVIEWER COMMENT	RESPONSE New or amended passages are shown in bold blue . Page and paragraph references correspond to the revised manuscript with tracked changes.
REVIEWER 2		
2.1	The paper "Carbon Footprint Distributions of Lithium-Ion Batteries and Their Materials" addresses a significant aspect of environmental impact assessment in the context of lithium-ion batteries (LIBs). The study focuses on the carbon footprint (CF) associated with LIBs, particularly the contributions from the sourcing and processing of key materials such as lithium, nickel, and cobalt. The paper employs a novel cost-based methodology to estimate emission curves for lithium, nickel, and cobalt, which allows for a more nuanced understanding of the CF associated with these materials. This approach adds some value by highlighting the variance in emissions based on different mining sites and processing methods. The paper is quite interesting. However:	We thank the reviewer for the time taken to thoroughly evaluate this manuscript. The recognition of the significance of our study is greatly appreciated. The reviewer's comments and references to existing work have been invaluable in revising the manuscript and in better highlighting areas for future research.
2.2	Some problems rely on proprietary data that cannot be shared, mainly the mining data. This makes it extremely hard to better understand and reproduce the analysis.	We acknowledge the importance of data reproducibility and open access in academia. Unfortunately, the seminal scope of our study (both breadth and depth) necessitated the use of proprietary data that we cannot disclose. To mitigate concerns regarding transparency and reproducibility, we made the following substantial improvements:  1. Our source files now include unique identifiers and URLs to all individual S&P Global Mine Economics assets, allowing anyone with access to the S&P Global Capital IQ Platform (available at many academic institutions) to retrieve the underlying cost data. 2. To ensure operability of our method and code for readers without access to the proprietary data bases, our revised repository now features additional open-access data sources for electricity prices for industrial consumers (statista.com and globalpetrolprices.com for BloombergNEF data), diesel prices (iea.org and globalpetrolprices.com for BloombergNEF data) and grid CO₂ intensities (ourworldindata.com for ecoinvent). While these sources provide insights similar to the

		proprietary data, they are less spatially detailed, which is why we rely on proprietary data for our analysis. 3. We have also added an Excel calculation file to the repository, substituting proprietary S&P Global data with illustrative data for two generic lithium mines. Using the open-access data sources from above (2), this new file allows the replication of our method for determining mine-level CFs. To deal with input data ranges and uncertainties, note that running the Monte Carlo simulation in Excel requires the @RISK add-on tool, which is available for a free 15-day trial. This Excel file contains all relevant CF contributors at the mine level, but the Python code provided in the repository (02a_Create_Emission_Curves.ipynb) must be executed to add refining emissions (if necessary) and to produce visualisations similar to Figure 1a. To address the reviewer’s concern, we added the following paragraph to our manuscript: page 16, line 428 The Excel files in the repository for this paper uniquely identify all mines analysed through S&P Global IDs, enabling readers with access to the S&P Global database to use these modelling files for reproduction or further analysis. Although this analysis primarily utilises proprietary data for electricity, diesel, and reagents prices, as well as grid CO2 intensities, the accompanying repository also contains open-access estimates for these data categories. It is important to note, however, that these open-access estimates are available at a lower spatial resolution and are sourced from various data providers.
2.3	Please reconsider the bibliography: BloombergNEF. Electric Vehicle Outlook 2022, can be substituted with the 2024 version (https://about.bnef.com/electric-vehicle-outlook/#download)	We thank the reviewer for pointing out the availability of the newer version of BloombergNEF's Electric Vehicle Outlook. The reason for citing the 2022 edition is because we don't have access to the 2024 version. To address this and ensure robustness, we have substantiated all statements previously supported by the BNEF 2022 report with additional references. For data points corroborated by the IEA Global EV Outlook 2024, these have been cited directly. page 12, line 280 The battery and mining industry is approaching a pivotal phase³⁶, with the majority of cell production and mining capacity to be built by 2035^{24, 39}. In instances where the IEA Global EV Outlook does not support the claim, we have referred to publicly accessible and up-to-date market research from December 2023 by the leading intelligence company Fastmarkets. page 2, line 62

		In general, the strong literature focus on NMC is problematic considering LFP market share projections of over 35% by 2030^{24, 25}.
2.4	In addition consider the following report: Global EV Outlook 2024 https://www.iea.org/reports/global-ev-outlook-2024. This is very interesting because it shows very similar results for LFP batteries. Then the basic data used for the report are affordable. Please comment.	We thank the reviewer for pointing us to this report. Please note that the IEA report uses a number of different sources and assumptions that make direct comparisons to our results challenging. The most notable differences are:  1. The IEA report uses a different functional unit. While they report battery carbon footprints on a battery pack level, our study presents values on a battery cell level. Battery packs are typically considered usable traction batteries that can be mounted directly to EVs, i.e., they contain, next to the battery cells, also packaging (typically aluminium and/or steel), battery management systems (semiconductors and copper), and sometimes also the cooling system. Thus, the CFs of IEA and this study have a different functional unit. 2. Given the different functional unit, the IEA report uses different gravimetric energy densities to convert CFs from a mass- to a kWh-basis. The pack design (layout, shape, and wiring of battery cells) and material procurement and engineering decisions influence the energy densities significantly. Because this work intentionally does not focus on the integration and package engineering aspect of EV traction batteries, we report CF values on the cell level. The IEA uses 0.165 kWh/kg_{pack} for NMC811 and 0.135 kWh/kg_{pack} for LFP while we use real-world energy densities from Schöberl et al. (2024): 0.25 kWh/kg_{cell} for NMC811 and 0.16 kWh/kg_{cell} for LFP. The decrease between cell and pack densities for LFP is typically much lower than for NMC811, as LFP cells are generally safer to operate and can thus be tighter packed. 3. The IEA report uses GREET as their underlying inventory while we use ecoinvent. Simply changing the background database (e.g., using GREET instead of ecoinvent) can already significantly influence the results. Furthermore, according to their sources on page 161 of their report, they use third-party market data and non-GREET energy estimates for battery cell production without further specifying what the data sources are used for. This makes it hard to retrace their modelling decision and screen their “basic data”. With these caveats in mind, we can convert the CFs of the IEA report to a mass-basis. Using the pack-level energy densities from Annexe B of the report, we obtain  ➔ IEA NMC811: $\sim 100 \text{ kg}_{\text{CO}_2}/\text{kWh} * 0.165 \text{ kWh}/\text{kg}_{\text{pack}} = \sim 16.5 \text{ kg}_{\text{CO}_2}/\text{kg}_{\text{pack}}$ ➔ IEA LFP: $\sim 70 \text{ kg}_{\text{CO}_2}/\text{kWh} * 0.135 \text{ kWh}/\text{kg}_{\text{pack}} = \sim 9.45 \text{ kg}_{\text{CO}_2}/\text{kg}_{\text{pack}}$. SI Section 2.3 of our paper displays the cell-level CFs on a mass-basis, namely  ➔ This work NMC811 (mean): 18.60 kg_{CO2}/kg_{cell} ➔ This work LFP (mean): 9.97 kg_{CO2}/kg_{cell}

		Note that, in general, on a mass-basis, one would expect cell-level CFs to be in the same range as pack-level CFs since most additional pack components are energy-intensive to produce as well (e.g., electronics, aluminium). While each additional pack component reduces the gravimetric energy density—since no active material is added that contributes to the battery’s capacity—the overall capacity-basis CF increases. This is consistent with the (mass-basis)ecoinvent activities for battery cells and packs. Accordingly, the pack-level CFs reported by IEA are in the same ballpark as the cell-level CFs reported in this work. In fact, one of the significant contributions of this work is the quantification of the CF distributions. As shown in this manuscript, the CF range of both LFP and NMC cells is large, and the (pack-level) values reported by IEA agree well with the distribution from our Monte-Carlo simulation. Note that due to significantly different gravimetric energy densities, the capacity-basis (vis-à-vis mass- basis) CFs between IEA’s pack-level CFs and this work’s cell-level CFs vary substantially (c.f., response to comment 2.5).
	Minor comments:	We greatly appreciate that the reviewer differentiated between major and minor comments, allowing us to direct our attention accordingly.
2.5	I’m quite surprised about the NMC811 results, which are lower than those expected. Check Global EV Outlook 2024 https://www.iea.org/reports/global-ev-outlook-2024 and justify.	We appreciate the reviewer's attention to the NMC811 results presented in our study. The lower carbon footprint (CF) for NMC811 reported in the cited IEA Global EV Outlook 2024 compared to our work can be largely attributed to the use of a substantially lower gravimetric energy density by the IEA to convert the mass-basis CF to the capacity-basis CF, the metric ultimately used in their report. This difference does not indicate a disagreement but reflects the use of different functional units: cell- vs. pack-level (see above). It is noteworthy that the variation in energy densities is less pronounced for LFP. For further background and detailed information, we refer the reviewer to our response to comment 2.4.
2.6	The paper assumes no regional material preferences among battery producers, which might oversimplify the complexities of global supply chains.	We appreciate the reviewer's attention to detail and for highlighting this aspect. In developing the research design and scope, we internally debated the extent to which we should model procurement choices and the supply chain from mine to cell producer. We opted to assume no regional material preferences because future supply chains are uncertain and we believe that the potential insights gained do not justify the additional complexity. This decision is supported by several considerations. Firstly, most battery materials are commercially traded commodities, making regionalisation of current trade somewhat meaningless. While there are calls for establishing markets for "green" commodities, it is uncertain when, or if, such markets will materialise. Secondly, direct mine-to-cell-producer

contracts often do not represent physical trades but can be used as legal or financial instruments for hedging. Lastly, as indicated in the IEA Global Critical Minerals Outlook 2024 (figure below), the refining market for lithium, cobalt, and graphite is heavily dominated by China, which significantly diminishes the potential insights from further regionalisation.

Geographical distribution of refined material production for key energy transition minerals in the base case, 2023-2040 Open

IEA. Licence: CC BY 4.0

IEA Global Critical Minerals Outlook 2024

Against this backdrop, we would like to point out that our study already substantially advances state-of-the-art research by incorporating, for the first time, spatially- and cell chemistry-specific announced battery production locations and accounting for these in global battery CF distributions.

2.7	Although the paper acknowledges the potential of battery recycling to reduce environmental impacts, it does not extensively explore this aspect. Including more detailed scenarios on the role of recycling in CF reduction could provide a more holistic view.	We thank the reviewer for emphasising the importance of recycling in the context of battery carbon footprints and agree with the general relevance of this aspect. However, our decision to exclude detailed recycling scenarios from the analysis was deliberate. This study quantifies global carbon footprint distributions of lithium-ion batteries (LIBs) and does not focus on individual battery producer choices. Current literature, including Huang et al. (2023, Fig. 16a, b, c), Watari et al. (2019, Fig. 8b, c), and the IEA Global Critical Minerals Outlook 2024, indicates that the availability of recycled materials will constitute only a small fraction of the total material needs for LIBs through to at least 2030. Given this situation, the role of recycling in the global distribution of CFs is considered negligible for the scope of this study. In light of the reviewer's comment, we refined the relevant passage in the manuscript and added references: page 14, line 368 [...] For long-term policy planning, it is crucial to consider the multi-faceted implications of recycling, ranging from environmental impacts over profitability and technological innovation to supply chain security and geopolitics, requiring future work. [...]
2.8	The focus on CF is crucial, but a more comprehensive environmental impact assessment including factors like water usage, land use change, and biodiversity loss could provide a more complete picture of the environmental footprint of LIBs.	We thank the reviewer for highlighting the importance of including broader environmental impacts such as water usage, land use change, and biodiversity loss in our assessment. We agree that these factors are vital for a comprehensive environmental footprint analysis of lithium-ion batteries. Our novel methodology, while seminal in its spatial resolution of battery CFs, does not directly translate to other environmental impacts. The reason for this is that costs used to derive the Dollar Emission Intensities (DEI) for every mine site cannot be converted with high certainty to e.g., biodiversity losses of mining through land use changes or water usage. We recognize the importance of these factors and revise our limitations and future research section accordingly: page 14, line 358 While this work proposes a novel cost-based methodology and contributes to the breadth and depth of CF analyses of LIB and their materials, there remains a pressing need for further research. As this methodology does not rely on time-consuming data collection from mining and refining company reports, it can be more easily scaled to additional commodities and panel datasets. Critically, other environmental impacts such as water usage, and biodiversity loss of mining through land use change should be quantified with similar granularity.^{43,51–53} Applying the

		proposed cost-based methodology to impacts like biodiversity is less straightforward than for global warming and thus warrants future work. [...]
	Remarks on code availability	
2.9	The results are not reproducible. Several data are not available	We refer to our responses to comment 2.2.

No.	REVIEWER COMMENT	RESPONSE New or amended passages are shown in bold blue . Page and paragraph references correspond to the revised manuscript with tracked changes.
REVIEWER 3		
3.1	In recent years, several countries around the world have issued policies and regulations to encourage the localization of the battery industry chain. This study develops a novel cost-based approach to estimate emission curves for the key battery materials based on mining cost data. The research is both interesting and meaningful. However, in my view, there are still some critical issues that need to be addressed.	We thank the reviewer for their positive remarks on the novelty and significance of our research. We are committed to addressing any concerns to enhance the quality and impact of our study. The reviewer's vigilance helped us to greatly improve our analysis, which we are grateful for.
3.2	Abstract. The analysis of the background is too long and lacks quantitative results. It is recommended that the author rewrites the Abstract.	We thank the reviewer for highlighting the need for a more concise and result-focused abstract. We have revised our abstract accordingly, making the background shorter and incorporating more quantitative results. The updated abstract now reads: Lithium-ion batteries (LIBs) are pivotal in climate change mitigation. While LIBs' own carbon footprint (CF) raises concerns, existing CF studies are scattered, hard to compare and largely overlook the relevance of battery materials. Here, we go beyond traditional CF analysis and develop a novel cost-based approach, estimating emission curves for battery materials lithium, nickel and cobalt based on mining cost data. Combining the emission curves with regionalised battery production announcements, we present the most representative CF distributions (5th, 50th, 95th percentile) for NMC811 (55, 72, 115 kgCO₂ kWh⁻¹) and LFP (54, 63, 68 kgCO₂ kWh⁻¹) cathode LIBs to date. Our findings reveal the significant impact of material sourcing over production location on CF, with nickel and lithium identified as major contributors to the CF and its variance. This research moves the field forward by offering a nuanced understanding of battery CFs, aiding in the design of decarbonisation policies and strategies.
3.3	Introduction. The greenhouse gas emissions on battery production, please consider: a) 10.1016/j.etrans.2022.100169 b) 10.1016/j.jclepro.2022.133342	We thank the reviewer for bringing these references to our attention. We have incorporated these studies into our manuscript to enrich the discussion on greenhouse gas emissions associated with battery production.

		page 2, line 30 The climate benefits of LIB-enabled products are evident^{2,3} but the production of battery materials⁴⁻⁷ and the subsequent LIB cell manufacturing⁸⁻¹⁰ contribute significantly to greenhouse gas (GHG) emissions [...] page 12, line 281 While materials from battery recycling are expected to reduce environmental damages, raw material extraction will need to provide the lion share of battery materials in the foreseeable future³⁹⁻⁴⁵.
3.4	Figure 1b, e, f is unclear and it is recommended to change the fill color. Additionally, it is suggested to supplement the x-axes in Figures 1c, f, and i.	We thank the reviewer for their comment and the opportunity to improve Figure 1. For panels b, e, and h, we have revised the fill colour from transparent grey to black and made the median line transparent to improve clarity. For panels c, f, and i, we have redesigned these panels entirely. Instead of a waterfall chart with an ambiguous x-axis, we now show a doughnut diagram that includes information on country, deposit type, and data coverage. We believe that these changes, prompted by the reviewer's suggestions, have significantly improved the readability and clarity of the figures.
3.5	The synthesized graphite produced in China is characterized by high carbon emission profiles, and it constitutes about two-thirds of the global output. Why are the median values from the database lower than the median values from the literature?	The reviewer raises an important point as the literature and database value situation is unsatisfactory. In panels b, e, h of our figures, we present data collected from peer-reviewed studies, grey literature, and databases that meet our quality standards. We refer the reviewer to our extensive literature review and its methodology, detailed in SI Section 3, which underpins our literature database and panels b, e, h. The reviewer's observation regarding the lower database values compared to literature values also applies to lithium and nickel. Generally, the discrepancy noted is indeed concerning and merely reflects the current state of research. Interestingly, through expert interviews, we learned that the emissions for synthetic graphite from China could indeed be up to twice as high as the values reported in the literature (and even more so for databases). However, we have not included these higher estimates in our study to maintain methodological consistency and to ensure reproducibility. SI page 25, line 426 While Panel a and b of SI Figure 3 show that the CF of synthetic graphite production ranges from around 6 to 14 kgCO2 kgGraphite-1, expert interviews revealed that the CF can reach up to 25 kgCO2 kgGraphite-1 in extreme cases. However, we do not incorporate estimates obtained from expert interviews into this work's CF database (SI Section 3) to maintain methodological consistency and ensure reproducibility.

		Nonetheless, this strong discrepancy is testament to the highly insufficient data coverage and calls for more comprehensive research on battery-grade graphite.
3.6	The synthesized graphite produced in China is characterized by high carbon emission profiles, and it constitutes about two-thirds of the global output. Why are the median database values lower than median values from the literature?	We refer to our response to comment 3.5.
3.7	In numerous literature sources, the proportion of active substances is approximately one-third (for example, Ciez and Whitacre, 2019, Nature Sustainability 2(2), 148-156). Why does this study find that active substances account for more than 50% of the carbon footprint?	We thank the reviewer for this insightful question. The higher proportion of active substances accounting for more than 50% of the carbon footprint in our study can be attributed primarily to the explicit and more realistic modelling of critical materials. Here, the median CF (per kg) of these materials is higher (Figure 1a, d, g) than those typically reported in the literature or databases (Figure 1b, e, h), naturally resulting in a higher relative contribution. In other words, it is not primarily driven by diverging assumptions on active material use per total battery weight. The divergence of our results from existing studies underscores the significance of our novel approach, integrating detailed material CF profiles into cell-level battery CFs. Further discrepancies may arise from differing modelling decisions, such as using GREET instead of ecoinvent as a background database and inventory, including variations in grid CO₂ intensity.
3.8	From Figure 2c, it can be observed that the contribution of nickel sulfate is 77.5%. The authors' conclusion that "the CF variance is virtually exclusively driven by nickel sulfate procurement" is overly absolute.	We appreciate the reviewer's observation regarding the contribution of nickel sulfate in Figure 2c. In light of the reviewer's comment, we have revised our wording to more accurately reflect the data and offer an explanation: page 6, line 176 Figure 2c shows that the CF variance is predominantly driven by nickel sulfate, whose tri-modal probability density function (SI Figure 2c) is characterised by the three plateaus in its emission curve (Figure 1d).
3.9	Line 203. The reviewer did not find Figure 3d in the manuscript.	We thank the reviewer for spotting the mistake mentioning of the non-existing Fig 3d in the text. This was a leftover from an earlier draft version of the manuscript, where the panel labelling was different. We have modified the relevant text section accordingly, but for the sake of brevity, we refrain from detailing the changes in this response document.
3.10	Figure 3a shows that the most LFP is produced in China. The cathode current collector commonly used	We appreciate the reviewer's detailed observation regarding the components contributing to the carbon footprint in Figure 3.

	in batteries is usually aluminum foil, which is produced in China through an energy-intensive and high-emission process of electrolytic aluminum, often powered by coal-electricity. Figure 3c reveals that the carbon footprint of LFP is primarily contributed by lithium carbonate, electricity, and graphite. However, the absence of a contribution from the high-emission electrolytic aluminum process is puzzling.	We would like to clarify that Figure 3c does not display the contribution to the absolute CF but to the variance in CF. The CF contribution from aluminium can be observed in Figure 3b, where it is accounted for as 11.6%, thus in line with the reviewer's comment. As described in the figure caption, the bar plot in panel C illustrates the variance contribution from parameters sampled in the Monte-Carlo simulation. Since our sampling is limited to emission curves shown in Figure 1 and production locations, only these parameters contribute to the variance in the sample.
3.11	Line 219-221. Please provide evidence for the statement "the shapes and data coverage of their emission curves vary significantly."	We thank the reviewer for bringing this unclear sentence to our attention. We clarified this sentence by referring to the manuscript figure: page 9, line 228 [...] Due to the different characteristics of each battery material, the shapes (Figure 1 a,d,g) and data coverage (Figure 1 c,f,i) of their emission curves vary significantly, making broad generalisations inappropriate. [...]
3.12	Line 234-236. Why did the authors choose mid-sized SUV electric vehicles with built-in NMC batteries? What is the chemical system of NMC batteries?	We selected mid-sized SUV electric vehicles as this vehicle type represents the most sold EV category across all EV-relevant regions, according to the IEA 2024 report (see Figure below). A mid-sized model was chosen over a large-sized one to yield more balanced results and avoid extremes.

		Breakdown of battery electric car sales in selected countries and regions by segment, 2018-2023 Open ↗ <small>IEA. Licence: CC BY 4.0</small> ● Small car ● Medium car ● Large car ● SUV ● Pick-up truck IEA (2024), Link Furthermore, we opted for high-density NMC (vis-à-vis LFP) cathode chemistries because they remain the predominant choice for these mid-sized models, as per the IEA 2024 findings. The chemical configuration of the traction battery in our study includes NMC811 battery cells. It is important to note that we have integrated these battery cells (the functional unit of Figures 2 and 3) to the battery pack level which is then installed in EVs. For this integration, we use data from ecoinvent, which in turn relies on open-access documents from Dai et al. (2017, 2018) from the Argonne National Laboratory. This step is detailed in the “NMC battery contribution” sheet of the “Breakeven_calculation_BEV_ICE.xlsx” Excel file available in our repository.
3.13	Line 273. What exactly is "the evidence"? Are there any referenced literature or reports? Please list them.	We appreciate the reviewer's request for clarification regarding "the evidence" mentioned in our manuscript. By "the evidence," we were referring to the regional probability density functions (coloured lines) illustrated in Figures 2a and 3a, which show considerable overlap. This overlap demonstrates that the production country of battery cells does not largely determine the cells'

		CF. We have clarified this point in the manuscript by adjusting the sentence in question and being more precise: page 12, line 288 From an environmental policy perspective, the similarities of regional CF profiles (coloured lines in Figure 2/3a) suggest [...]
3.14	The emissions method proposed in the abstract of this paper, which is associated with cost, is barely addressed in the discuss section.	We appreciate the reviewer's observation regarding the coverage of the cost-based methodology in the discussion section and agree that both the contributions of this approach and its limitations warrant more extensive discussion. We have modified the relevant text passages in the discussion section accordingly to provide a more comprehensive analysis: page 14, line 358 While this work proposes a novel cost-based methodology and contributes to the breadth and depth of CF analyses of LIB and their materials, there remains a pressing need for further research. As this methodology does not rely on time-consuming data collection from mining and refining company reports, it can be more easily scaled to additional commodities and panel datasets. Critically, other environmental impacts such as water usage, and biodiversity loss of mining through land use change should be quantified with similar granularity.^{43,51-53} Applying the proposed cost-based methodology to impacts like biodiversity is less straightforward than for global warming and thus warrants future work. [...]
3.15	Line 294-297. Without considering upstream materials, "the CF profiles of Chinese-made LIBs largely coincide with those manufactured in the US or Europe" seems to differ from current research.	We are grateful for the reviewer's correct contextualisation of our findings, particularly in highlighting the discrepancy between the general notion in the field and our novel insights. We attribute this discrepancy to two primary factors: (1) the predominant influence of material choices over production locations, and (2) the reliance on national averages for grid CO2 intensities in most existing comparisons, rather than using more spatially explicit data. Notably, certain regions within China exhibit relatively low grid CO2 intensities, as illustrated in Figure 4b. Without accounting for this, it could lead to incorrect assumptions that all Chinese batteries inherently have a higher CF than those produced in the US or Europe. We have amended the text section to reflect this discussion and incorporate the reviewer's observation more clearly. page 12, line 311 [...] With the caveat of no regional materials preferences of battery producers, the CF profiles of Chinese-made LIBs largely coincide with those manufactured in the US or

		Europe. This new insight can be explained with materials' CF (variance) influence and the spatially-explicit grid CO2 intensities beyond national averages. [...]
3.16	Line 323-324. Please confirm whether the "low-carbon lithium carbonate from..." mentioned is indeed represented in Figure 4b. Verify the entire text and make any necessary revisions.	We thank the reviewer for spotting this mistake. We were, in fact, referring to Figure 4c. Necessary revisions have been made to ensure accuracy throughout the text, although – for the sake of brevity – these changes are not detailed in this response document.
3.17	Line 344-346. Nickel is also not LIB-specific and used in much larger quantities in stainless steels. Why does the author focus on studying it?	This is an important discussion point – we are grateful that this came up during the review. The inclusion of nickel in this study is justified for two main reasons: Firstly, the carbon footprint impact of nickel in lithium-ion batteries (LIBs) is substantial. Therefore, even if the entire nickel supply chain were not specific to LIBs, it would still be crucial to include it due to its environmental significance. This differs from the materials mentioned in line 379f. Secondly, our study focuses exclusively on nickel that could theoretically be processed into high-purity nickel sulfate, which is used in LIBs. The processing routes for nickel chemicals, such as nickel sulfate, differ fundamentally from those used for stainless steel, thereby making it LIB-specific and justifying a detailed analysis. For a deeper understanding of how we delineate the nickel supply chain for this purpose, please refer to SI Section 1.7. To address the reviewer's comment and increase clarity, we added the following sentence to the manuscript: page 15, line 379 The battery material selection is based on CF contribution and LIB-specificity. Copper, aluminium, phosphate, manganese, and iron (sulfate) are not LIB-specific and are used in much larger quantities in other sectors. The production routes for Nickel chemicals used in LIBs differ fundamentally from those in the steel industry, the largest Nickel consumer³¹. Furthermore, manganese, and iron (sulfate) have a negligible CF contribution as shown in Figure 3b. [...]
3.18	Line 452-454. It is suggested that the content of "expert interviews and educated guesses" be supplemented to enhance credibility.	We thank the reviewer for directing our attention to the need for enhanced credibility regarding the use of "expert interviews and educated guesses." Following the reviewer's suggestion, we have clarified the sentence and made the number of estimates explicit (previously "educated guesses"). The revised section now reads: page 19, line 497 [...] For all input parameters, we define uncertainty distributions based on expert interviews, own simulations or multiple data sources. Where only single data points were

		available, we assumed triangular distributions with +-5% (diesel prices) and +-10% (reagent prices). [...]
3.19	Line 567-578. Please provide the data on the provincial- (CN), state- (US), and country-level (Europe, JP, KR) understanding of announced LIB production capacities of NMC and LFP until 2030.	We appreciate the reviewer's request for detailed data on provincial (CN), state (US), and country-level (Europe, JP, KR) LIB production capacities. Unfortunately, we are unable to disclose proprietary data from BloombergNEF (BNEF), which is our primary data source for this. However, the BNEF data aligns closely with Table S3 of Kallitsis et al., who collated open-source battery manufacturing capacity announcements. BNEF uses a consistent methodology across all regions, and some of their data points include specific information on battery chemistry (LFP vs. NMC). For consistency and reliability, we have chosen BNEF as our main data source, corroborating it as necessary, as detailed in the section "Collating Regional Announced Battery Production Capacity" of our manuscript. Additionally, to ensure operability of our method and code for readers without access to the proprietary data bases, our revised repository now features additional open-access data sources for electricity prices for industrial consumers (statista.com and globalpetrolprices.com for BloombergNEF data), diesel prices (iea.org and globalpetrolprices.com for BloombergNEF data) and grid CO2 intensities (ourworldindata.com for ecoinvent). While these sources provide insights similar to the proprietary data, they are less spatially detailed, which is why we rely on proprietary data for our analysis. It is important to note that in our analysis, the absolute capacity numbers do not shape the results as much as the capacity ratios do. Within regions, different capacity ratios of provinces (CN), states (US), and countries (Europe, JP, KR) alongside their local carbon grid intensities determine the shape of the regional probability density functions (coloured lines in Figures 2/3a) and the distribution of the regional jitter plot (coloured dots in Figures 2/3a). Capacity ratios between regions are crucial for the global CF distribution (black line in Figures 2/3a) and the global jitter plot (black dots in Figures 2/3a). Note that, for this analysis, we use tracked battery announcements that do not necessarily reflect actual capacity build-up by 2030.

REVIEWERS' COMMENTS

Reviewer #2 (Remarks to the Author):

satisfied with the revision. The only problem concerns the data availability because some of these are proprietary data.

Reviewer #3 (Remarks to the Author):

The author has effectively addressed all the issues I raised.

No.	REVIEWER COMMENT	RESPONSE New or amended passages are shown in bold blue . Page and paragraph references correspond to the revised manuscript with tracked changes.
REVIEWER 1		
1.1	This research provides results on carbon footprints of different salts for lithium ion battery. It is interesting to combine cost into the carbon footprints analysis. The reviewer has the following comments:	We thank the reviewer for their time and insightful feedback on our manuscript. We value their comments as they help us enhance the clarity and impact of our research.
1.2	It is necessary to clearly explain Fig. 1. The results seem not correct referring to the calculation raw data. Why the CF is low for CL while it is high for CN concerning lithium carbonate for instance? The source data are not convincing. They need to be verified.	We gladly clarify the methodology for deriving the carbon footprints (CF) shown in Figure 1. Generally, the differences in CFs of mines can be explained by walking through the formulas provided in the Method section of the manuscript. For example, as explained in Formula (1), the CF for each mine is calculated as the sum of individual CF contributions from various cost categories, such as electricity, diesel, or chemical reagents. Each contribution is influenced by the mine-specific cost (e.g., 100 USD spent on diesel per 1 kg of lithium carbonate) and the “Dollar Emission Intensity” (DEI) (e.g., 1 kg of CO₂ per 1 USD spent on diesel). These costs and DEI values are typically specific for each mine and cost category. Additionally, where mine-specific data from S&P Global only covers part of the processing chain (e.g., up to lithium concentrate for spodumene-mined lithium), we have incorporated emissions from refining to battery-grade material using literature values. We refer to the method section and the attached Supplementary Information (SI) for detailed explanations. Regarding the data sources and their verification, we acknowledge the importance of data reproducibility and open access in academia. Unfortunately, the seminal scope of our study (both breadth and depth) necessitated the use of proprietary data that we cannot disclose. To mitigate concerns regarding transparency and reproducibility, we have implemented the following changes:  1. Our source files now include unique identifiers and URLs to all individual S&P Global Mine Economics assets, allowing anyone with access to the S&P Global Capital IQ Platform (available at many academic institutions) to retrieve the underlying cost data. 2. To ensure operability of our method and code for readers without access to the proprietary data bases, our revised repository now features additional open-access data sources for electricity prices for industrial consumers (statista.com and globalpetrolprices.com for BloombergNEF data), diesel prices (iea.org and

		globalpetrolprices.com for BloombergNEF data) and grid CO2 intensities (ourworldindata.com for ecoinvent). While these sources provide insights similar to the proprietary data, they are less spatially detailed, which is why we rely on proprietary data for our analysis. 3. We have also added an Excel calculation file to the repository, substituting proprietary S&P Global data with illustrative data for two generic lithium mines. Using the open-access data sources from above (2), this new file allows the replication of our method for determining mine-level CFs. To deal with input data ranges and uncertainties, the Monte Carlo simulation in Excel requires the @RISK add-on tool, which is available for a free 15-day trial. This Excel file contains all relevant CF contributors at the mine level, but the Python code provided in the repository (02a_Create_Emission_Curves.ipynb) must be executed to add refining emissions (if necessary) and to produce visualisations similar to Figure 1a. In terms of data reliability, our source data from S&P Global is extensively used by industry and has been validated in expert interviews. Furthermore, the CF values determined using our novel methodology align well with conventional mine-specific CF estimates. Additionally, in the uncertainty calculations for the CF of battery materials (Figure 1), we apply a $\pm 10\%$ triangular distribution to all data used from S&P Global (see SI Section 1.5). To address the reviewer’s concern, we added the following paragraph to our manuscript: page 16, line 428 The Excel files in the repository for this paper uniquely identify all mines analysed through S&P Global IDs, enabling readers with access to the S&P Global database to use these modelling files for reproduction or further analysis. Although this analysis primarily utilises proprietary data for electricity, diesel, and reagents prices, as well as grid CO2 intensities, the accompanying repository also contains open-access estimates for these data categories. It is important to note, however, that these open-access estimates are available at a lower spatial resolution and are sourced from various data providers.
1.3.1	Figure 2 is also not correct. Concerning NCM811 the production rate given in the figure is not precise and the data can vary. For instance, the NCM811 materials can be produced in one country while the battery is produced in another.	We thank the reviewer for their attention to the details in Figure 2. The production shares are indeed not “precise” but intentionally represented as probabilistic, thus reflecting the inherent diversity in production locations. These values are derived from a Monte-Carlo simulation involving over 11,000 runs, grounded in proprietary Bloomberg NEF data and corroborated by the findings of Kallitsis et al. (2024), as discussed in the Collating regional announced battery production capacity method section of our manuscript.

		We appreciate the reviewer's concern regarding the geographical discrepancies between the production of NCM811 materials and the assembly of battery cells. This work's research design explicitly accounts for the reviewer's remark: in the Monte-Carlo simulation, we not only vary the location of the cell manufacturing site (paragraph above) but also the provenance of the material. Specifically, we build on Figure 1 to derive probability density functions (SI Figure 2) that are used as underlying material distributions for the Monte Carlo algorithm. This approach ensures a robust analysis of global production dynamics.
1.3.2	Where is the electrolyte?	The carbon footprint contribution of the electrolyte, quantified at 3.24 kgCO₂/kWh, is indeed included in our analysis but aggregated under the "Cell Other [10.8 %]" category in Figure 2b for clarity and visual succinctness (resp. [4.6 %] in Figure 3b). For a detailed breakdown of each component's carbon footprint, including the electrolyte, the reviewer is referred to SI Table 4 where we present the disaggregated data.
1.4.1	Figure 2b the share of electricity has two parts. What are the difference?	We thank the reviewer for highlighting this. The separation of different electricity contributions in Figure 2b aims to illustrate the distinct value chain steps where electricity is consumed. Notably, a significant portion of electricity is required for producing the cathode, especially during the calcination of NMC hydroxide to NMC oxide. In the initial submission, different colour schemes were used to represent cell-level (blue), cathode (lilac), and anode (brown) processes. However, a colour legend was missing in the Figure panel. To address the reviewer's helpful comment, a colour legend has now been added to enhance clarity in revised Figure 2 (and Figure 3 and SI Figures 7, 8 ,9).
1.4.2	in Fig. 2c, the share of contribute to variance for NiSO ₄ dominates. It is necessary to provide reasons and how much the divation will be.	We thank the reviewer for highlighting that the dominant contribution of nickel sulphate to the overall variance in Fig. 2c was not clearly explained in our initial submission. The mechanistic reason for this dominance is that the underlying probability density function for nickel sulphate is tri-modal over a large domain, as shown in Nickel incl. laterite in SI Figure 2 (third column). Since variance is calculated by considering the squared differences between the dataset's mean and its individual data points, a distribution with pronounced outliers will lead to large variance. SI Table 6 contains numerical results for variances shown in the main manuscript. Following the reviewer's recommendation, we have added the following explanation: page 6, line 176 Figure 2c shows that the CF variance is predominantly driven by nickel sulfate, whose tri-modal probability density function (SI Figure 2c) is characterised by the three plateaus in its emission curve (Figure 1d).

1.5	Fig. 3 has the same problem.	We agree with the reviewer's helpful comment. In light of our responses to comments 1.4.1 and 1.4.2, we have made analogous amendments to Figure 3 as well.
-----	------------------------------	---

No.	REVIEWER COMMENT	RESPONSE New or amended passages are shown in bold blue . Page and paragraph references correspond to the revised manuscript with tracked changes.
REVIEWER 2		
2.1	The paper "Carbon Footprint Distributions of Lithium-Ion Batteries and Their Materials" addresses a significant aspect of environmental impact assessment in the context of lithium-ion batteries (LIBs). The study focuses on the carbon footprint (CF) associated with LIBs, particularly the contributions from the sourcing and processing of key materials such as lithium, nickel, and cobalt. The paper employs a novel cost-based methodology to estimate emission curves for lithium, nickel, and cobalt, which allows for a more nuanced understanding of the CF associated with these materials. This approach adds some value by highlighting the variance in emissions based on different mining sites and processing methods. The paper is quite interesting. However:	We thank the reviewer for the time taken to thoroughly evaluate this manuscript. The recognition of the significance of our study is greatly appreciated. The reviewer's comments and references to existing work have been invaluable in revising the manuscript and in better highlighting areas for future research.
2.2	Some problems rely on proprietary data that cannot be shared, mainly the mining data. This makes it extremely hard to better understand and reproduce the analysis.	We acknowledge the importance of data reproducibility and open access in academia. Unfortunately, the seminal scope of our study (both breadth and depth) necessitated the use of proprietary data that we cannot disclose. To mitigate concerns regarding transparency and reproducibility, we made the following substantial improvements:  1. Our source files now include unique identifiers and URLs to all individual S&P Global Mine Economics assets, allowing anyone with access to the S&P Global Capital IQ Platform (available at many academic institutions) to retrieve the underlying cost data. 2. To ensure operability of our method and code for readers without access to the proprietary data bases, our revised repository now features additional open-access data sources for electricity prices for industrial consumers (statista.com and globalpetrolprices.com for BloombergNEF data), diesel prices (iea.org and globalpetrolprices.com for BloombergNEF data) and grid CO₂ intensities (ourworldindata.com for ecoinvent). While these sources provide insights similar to the

		proprietary data, they are less spatially detailed, which is why we rely on proprietary data for our analysis. 3. We have also added an Excel calculation file to the repository, substituting proprietary S&P Global data with illustrative data for two generic lithium mines. Using the open-access data sources from above (2), this new file allows the replication of our method for determining mine-level CFs. To deal with input data ranges and uncertainties, note that running the Monte Carlo simulation in Excel requires the @RISK add-on tool, which is available for a free 15-day trial. This Excel file contains all relevant CF contributors at the mine level, but the Python code provided in the repository (02a_Create_Emission_Curves.ipynb) must be executed to add refining emissions (if necessary) and to produce visualisations similar to Figure 1a. To address the reviewer’s concern, we added the following paragraph to our manuscript: page 16, line 428 The Excel files in the repository for this paper uniquely identify all mines analysed through S&P Global IDs, enabling readers with access to the S&P Global database to use these modelling files for reproduction or further analysis. Although this analysis primarily utilises proprietary data for electricity, diesel, and reagents prices, as well as grid CO2 intensities, the accompanying repository also contains open-access estimates for these data categories. It is important to note, however, that these open-access estimates are available at a lower spatial resolution and are sourced from various data providers.
2.3	Please reconsider the bibliography: BloombergNEF. Electric Vehicle Outlook 2022, can be substituted with the 2024 version (https://about.bnef.com/electric-vehicle-outlook/#download)	We thank the reviewer for pointing out the availability of the newer version of BloombergNEF's Electric Vehicle Outlook. The reason for citing the 2022 edition is because we don't have access to the 2024 version. To address this and ensure robustness, we have substantiated all statements previously supported by the BNEF 2022 report with additional references. For data points corroborated by the IEA Global EV Outlook 2024, these have been cited directly. page 12, line 280 The battery and mining industry is approaching a pivotal phase³⁶, with the majority of cell production and mining capacity to be built by 2035^{24, 39}. In instances where the IEA Global EV Outlook does not support the claim, we have referred to publicly accessible and up-to-date market research from December 2023 by the leading intelligence company Fastmarkets. page 2, line 62

		In general, the strong literature focus on NMC is problematic considering LFP market share projections of over 35% by 2030^{24, 25}.
2.4	In addition consider the following report: Global EV Outlook 2024 https://www.iea.org/reports/global-ev-outlook-2024. This is very interesting because it shows very similar results for LFP batteries. Then the basic data used for the report are affordable. Please comment.	We thank the reviewer for pointing us to this report. Please note that the IEA report uses a number of different sources and assumptions that make direct comparisons to our results challenging. The most notable differences are:  1. The IEA report uses a different functional unit. While they report battery carbon footprints on a battery pack level, our study presents values on a battery cell level. Battery packs are typically considered usable traction batteries that can be mounted directly to EVs, i.e., they contain, next to the battery cells, also packaging (typically aluminium and/or steel), battery management systems (semiconductors and copper), and sometimes also the cooling system. Thus, the CFs of IEA and this study have a different functional unit. 2. Given the different functional unit, the IEA report uses different gravimetric energy densities to convert CFs from a mass- to a kWh-basis. The pack design (layout, shape, and wiring of battery cells) and material procurement and engineering decisions influence the energy densities significantly. Because this work intentionally does not focus on the integration and package engineering aspect of EV traction batteries, we report CF values on the cell level. The IEA uses 0.165 kWh/kg_{pack} for NMC811 and 0.135 kWh/kg_{pack} for LFP while we use real-world energy densities from Schöberl et al. (2024): 0.25 kWh/kg_{cell} for NMC811 and 0.16 kWh/kg_{cell} for LFP. The decrease between cell and pack densities for LFP is typically much lower than for NMC811, as LFP cells are generally safer to operate and can thus be tighter packed. 3. The IEA report uses GREET as their underlying inventory while we use ecoinvent. Simply changing the background database (e.g., using GREET instead of ecoinvent) can already significantly influence the results. Furthermore, according to their sources on page 161 of their report, they use third-party market data and non-GREET energy estimates for battery cell production without further specifying what the data sources are used for. This makes it hard to retrace their modelling decision and screen their “basic data”. With these caveats in mind, we can convert the CFs of the IEA report to a mass-basis. Using the pack-level energy densities from Annexe B of the report, we obtain  ➔ IEA NMC811: $\sim 100 \text{ kg}_{\text{CO}_2}/\text{kWh} * 0.165 \text{ kWh}/\text{kg}_{\text{pack}} = \sim 16.5 \text{ kg}_{\text{CO}_2}/\text{kg}_{\text{pack}}$ ➔ IEA LFP: $\sim 70 \text{ kg}_{\text{CO}_2}/\text{kWh} * 0.135 \text{ kWh}/\text{kg}_{\text{pack}} = \sim 9.45 \text{ kg}_{\text{CO}_2}/\text{kg}_{\text{pack}}$. SI Section 2.3 of our paper displays the cell-level CFs on a mass-basis, namely  ➔ This work NMC811 (mean): 18.60 kg_{CO2}/kg_{cell} ➔ This work LFP (mean): 9.97 kg_{CO2}/kg_{cell}

		Note that, in general, on a mass-basis, one would expect cell-level CFs to be in the same range as pack-level CFs since most additional pack components are energy-intensive to produce as well (e.g., electronics, aluminium). While each additional pack component reduces the gravimetric energy density—since no active material is added that contributes to the battery’s capacity—the overall capacity-basis CF increases. This is consistent with the (mass-basis)ecoinvent activities for battery cells and packs. Accordingly, the pack-level CFs reported by IEA are in the same ballpark as the cell-level CFs reported in this work. In fact, one of the significant contributions of this work is the quantification of the CF distributions. As shown in this manuscript, the CF range of both LFP and NMC cells is large, and the (pack-level) values reported by IEA agree well with the distribution from our Monte-Carlo simulation. Note that due to significantly different gravimetric energy densities, the capacity-basis (vis-à-vis mass- basis) CFs between IEA’s pack-level CFs and this work’s cell-level CFs vary substantially (c.f., response to comment 2.5).
	Minor comments:	We greatly appreciate that the reviewer differentiated between major and minor comments, allowing us to direct our attention accordingly.
2.5	I’m quite surprised about the NMC811 results, which are lower than those expected. Check Global EV Outlook 2024 https://www.iea.org/reports/global-ev-outlook-2024 and justify.	We appreciate the reviewer's attention to the NMC811 results presented in our study. The lower carbon footprint (CF) for NMC811 reported in the cited IEA Global EV Outlook 2024 compared to our work can be largely attributed to the use of a substantially lower gravimetric energy density by the IEA to convert the mass-basis CF to the capacity-basis CF, the metric ultimately used in their report. This difference does not indicate a disagreement but reflects the use of different functional units: cell- vs. pack-level (see above). It is noteworthy that the variation in energy densities is less pronounced for LFP. For further background and detailed information, we refer the reviewer to our response to comment 2.4.
2.6	The paper assumes no regional material preferences among battery producers, which might oversimplify the complexities of global supply chains.	We appreciate the reviewer's attention to detail and for highlighting this aspect. In developing the research design and scope, we internally debated the extent to which we should model procurement choices and the supply chain from mine to cell producer. We opted to assume no regional material preferences because future supply chains are uncertain and we believe that the potential insights gained do not justify the additional complexity. This decision is supported by several considerations. Firstly, most battery materials are commercially traded commodities, making regionalisation of current trade somewhat meaningless. While there are calls for establishing markets for "green" commodities, it is uncertain when, or if, such markets will materialise. Secondly, direct mine-to-cell-producer

contracts often do not represent physical trades but can be used as legal or financial instruments for hedging. Lastly, as indicated in the IEA Global Critical Minerals Outlook 2024 (figure below), the refining market for lithium, cobalt, and graphite is heavily dominated by China, which significantly diminishes the potential insights from further regionalisation.

Geographical distribution of refined material production for key energy transition minerals in the base case, 2023-2040 Open

IEA. Licence: CC BY 4.0

IEA Global Critical Minerals Outlook 2024

Against this backdrop, we would like to point out that our study already substantially advances state-of-the-art research by incorporating, for the first time, spatially- and cell chemistry-specific announced battery production locations and accounting for these in global battery CF distributions.

2.7	Although the paper acknowledges the potential of battery recycling to reduce environmental impacts, it does not extensively explore this aspect. Including more detailed scenarios on the role of recycling in CF reduction could provide a more holistic view.	We thank the reviewer for emphasising the importance of recycling in the context of battery carbon footprints and agree with the general relevance of this aspect. However, our decision to exclude detailed recycling scenarios from the analysis was deliberate. This study quantifies global carbon footprint distributions of lithium-ion batteries (LIBs) and does not focus on individual battery producer choices. Current literature, including Huang et al. (2023, Fig. 16a, b, c), Watari et al. (2019, Fig. 8b, c), and the IEA Global Critical Minerals Outlook 2024, indicates that the availability of recycled materials will constitute only a small fraction of the total material needs for LIBs through to at least 2030. Given this situation, the role of recycling in the global distribution of CFs is considered negligible for the scope of this study. In light of the reviewer's comment, we refined the relevant passage in the manuscript and added references: page 14, line 368 [...] For long-term policy planning, it is crucial to consider the multi-faceted implications of recycling, ranging from environmental impacts over profitability and technological innovation to supply chain security and geopolitics, requiring future work. [...]
2.8	The focus on CF is crucial, but a more comprehensive environmental impact assessment including factors like water usage, land use change, and biodiversity loss could provide a more complete picture of the environmental footprint of LIBs.	We thank the reviewer for highlighting the importance of including broader environmental impacts such as water usage, land use change, and biodiversity loss in our assessment. We agree that these factors are vital for a comprehensive environmental footprint analysis of lithium-ion batteries. Our novel methodology, while seminal in its spatial resolution of battery CFs, does not directly translate to other environmental impacts. The reason for this is that costs used to derive the Dollar Emission Intensities (DEI) for every mine site cannot be converted with high certainty to e.g., biodiversity losses of mining through land use changes or water usage. We recognize the importance of these factors and revise our limitations and future research section accordingly: page 14, line 358 While this work proposes a novel cost-based methodology and contributes to the breadth and depth of CF analyses of LIB and their materials, there remains a pressing need for further research. As this methodology does not rely on time-consuming data collection from mining and refining company reports, it can be more easily scaled to additional commodities and panel datasets. Critically, other environmental impacts such as water usage, and biodiversity loss of mining through land use change should be quantified with similar granularity.^{43,51–53} Applying the

		proposed cost-based methodology to impacts like biodiversity is less straightforward than for global warming and thus warrants future work. [...]
	Remarks on code availability	
2.9	The results are not reproducible. Several data are not available	We refer to our responses to comment 2.2.

No.	REVIEWER COMMENT	RESPONSE New or amended passages are shown in bold blue . Page and paragraph references correspond to the revised manuscript with tracked changes.
REVIEWER 3		
3.1	In recent years, several countries around the world have issued policies and regulations to encourage the localization of the battery industry chain. This study develops a novel cost-based approach to estimate emission curves for the key battery materials based on mining cost data. The research is both interesting and meaningful. However, in my view, there are still some critical issues that need to be addressed.	We thank the reviewer for their positive remarks on the novelty and significance of our research. We are committed to addressing any concerns to enhance the quality and impact of our study. The reviewer's vigilance helped us to greatly improve our analysis, which we are grateful for.
3.2	Abstract. The analysis of the background is too long and lacks quantitative results. It is recommended that the author rewrites the Abstract.	We thank the reviewer for highlighting the need for a more concise and result-focused abstract. We have revised our abstract accordingly, making the background shorter and incorporating more quantitative results. The updated abstract now reads: Lithium-ion batteries (LIBs) are pivotal in climate change mitigation. While LIBs' own carbon footprint (CF) raises concerns, existing CF studies are scattered, hard to compare and largely overlook the relevance of battery materials. Here, we go beyond traditional CF analysis and develop a novel cost-based approach, estimating emission curves for battery materials lithium, nickel and cobalt based on mining cost data. Combining the emission curves with regionalised battery production announcements, we present the most representative CF distributions (5th, 50th, 95th percentile) for NMC811 (55, 72, 115 kgCO₂ kWh⁻¹) and LFP (54, 63, 68 kgCO₂ kWh⁻¹) cathode LIBs to date. Our findings reveal the significant impact of material sourcing over production location on CF, with nickel and lithium identified as major contributors to the CF and its variance. This research moves the field forward by offering a nuanced understanding of battery CFs, aiding in the design of decarbonisation policies and strategies.
3.3	Introduction. The greenhouse gas emissions on battery production, please consider: a) 10.1016/j.etrans.2022.100169 b) 10.1016/j.jclepro.2022.133342	We thank the reviewer for bringing these references to our attention. We have incorporated these studies into our manuscript to enrich the discussion on greenhouse gas emissions associated with battery production.

		page 2, line 30 The climate benefits of LIB-enabled products are evident^{2,3} but the production of battery materials⁴⁻⁷ and the subsequent LIB cell manufacturing⁸⁻¹⁰ contribute significantly to greenhouse gas (GHG) emissions [...] page 12, line 281 While materials from battery recycling are expected to reduce environmental damages, raw material extraction will need to provide the lion share of battery materials in the foreseeable future³⁹⁻⁴⁵.
3.4	Figure 1b, e, f is unclear and it is recommended to change the fill color. Additionally, it is suggested to supplement the x-axes in Figures 1c, f, and i.	We thank the reviewer for their comment and the opportunity to improve Figure 1. For panels b, e, and h, we have revised the fill colour from transparent grey to black and made the median line transparent to improve clarity. For panels c, f, and i, we have redesigned these panels entirely. Instead of a waterfall chart with an ambiguous x-axis, we now show a doughnut diagram that includes information on country, deposit type, and data coverage. We believe that these changes, prompted by the reviewer's suggestions, have significantly improved the readability and clarity of the figures.
3.5	The synthesized graphite produced in China is characterized by high carbon emission profiles, and it constitutes about two-thirds of the global output. Why are the median values from the database lower than the median values from the literature?	The reviewer raises an important point as the literature and database value situation is unsatisfactory. In panels b, e, h of our figures, we present data collected from peer-reviewed studies, grey literature, and databases that meet our quality standards. We refer the reviewer to our extensive literature review and its methodology, detailed in SI Section 3, which underpins our literature database and panels b, e, h. The reviewer's observation regarding the lower database values compared to literature values also applies to lithium and nickel. Generally, the discrepancy noted is indeed concerning and merely reflects the current state of research. Interestingly, through expert interviews, we learned that the emissions for synthetic graphite from China could indeed be up to twice as high as the values reported in the literature (and even more so for databases). However, we have not included these higher estimates in our study to maintain methodological consistency and to ensure reproducibility. SI page 25, line 426 While Panel a and b of SI Figure 3 show that the CF of synthetic graphite production ranges from around 6 to 14 kgCO2 kgGraphite-1, expert interviews revealed that the CF can reach up to 25 kgCO2 kgGraphite-1 in extreme cases. However, we do not incorporate estimates obtained from expert interviews into this work's CF database (SI Section 3) to maintain methodological consistency and ensure reproducibility.

		Nonetheless, this strong discrepancy is testament to the highly insufficient data coverage and calls for more comprehensive research on battery-grade graphite.
3.6	The synthesized graphite produced in China is characterized by high carbon emission profiles, and it constitutes about two-thirds of the global output. Why are the median database values lower than median values from the literature?	We refer to our response to comment 3.5.
3.7	In numerous literature sources, the proportion of active substances is approximately one-third (for example, Ciez and Whitacre, 2019, Nature Sustainability 2(2), 148-156). Why does this study find that active substances account for more than 50% of the carbon footprint?	We thank the reviewer for this insightful question. The higher proportion of active substances accounting for more than 50% of the carbon footprint in our study can be attributed primarily to the explicit and more realistic modelling of critical materials. Here, the median CF (per kg) of these materials is higher (Figure 1a, d, g) than those typically reported in the literature or databases (Figure 1b, e, h), naturally resulting in a higher relative contribution. In other words, it is not primarily driven by diverging assumptions on active material use per total battery weight. The divergence of our results from existing studies underscores the significance of our novel approach, integrating detailed material CF profiles into cell-level battery CFs. Further discrepancies may arise from differing modelling decisions, such as using GREET instead of ecoinvent as a background database and inventory, including variations in grid CO₂ intensity.
3.8	From Figure 2c, it can be observed that the contribution of nickel sulfate is 77.5%. The authors' conclusion that "the CF variance is virtually exclusively driven by nickel sulfate procurement" is overly absolute.	We appreciate the reviewer's observation regarding the contribution of nickel sulfate in Figure 2c. In light of the reviewer's comment, we have revised our wording to more accurately reflect the data and offer an explanation: page 6, line 176 Figure 2c shows that the CF variance is predominantly driven by nickel sulfate, whose tri-modal probability density function (SI Figure 2c) is characterised by the three plateaus in its emission curve (Figure 1d).
3.9	Line 203. The reviewer did not find Figure 3d in the manuscript.	We thank the reviewer for spotting the mistake mentioning of the non-existing Fig 3d in the text. This was a leftover from an earlier draft version of the manuscript, where the panel labelling was different. We have modified the relevant text section accordingly, but for the sake of brevity, we refrain from detailing the changes in this response document.
3.10	Figure 3a shows that the most LFP is produced in China. The cathode current collector commonly used	We appreciate the reviewer's detailed observation regarding the components contributing to the carbon footprint in Figure 3.

	in batteries is usually aluminum foil, which is produced in China through an energy-intensive and high-emission process of electrolytic aluminum, often powered by coal-electricity. Figure 3c reveals that the carbon footprint of LFP is primarily contributed by lithium carbonate, electricity, and graphite. However, the absence of a contribution from the high-emission electrolytic aluminum process is puzzling.	We would like to clarify that Figure 3c does not display the contribution to the absolute CF but to the variance in CF. The CF contribution from aluminium can be observed in Figure 3b, where it is accounted for as 11.6%, thus in line with the reviewer's comment. As described in the figure caption, the bar plot in panel C illustrates the variance contribution from parameters sampled in the Monte-Carlo simulation. Since our sampling is limited to emission curves shown in Figure 1 and production locations, only these parameters contribute to the variance in the sample.
3.11	Line 219-221. Please provide evidence for the statement "the shapes and data coverage of their emission curves vary significantly."	We thank the reviewer for bringing this unclear sentence to our attention. We clarified this sentence by referring to the manuscript figure: page 9, line 228 [...] Due to the different characteristics of each battery material, the shapes (Figure 1 a,d,g) and data coverage (Figure 1 c,f,i) of their emission curves vary significantly, making broad generalisations inappropriate. [...]
3.12	Line 234-236. Why did the authors choose mid-sized SUV electric vehicles with built-in NMC batteries? What is the chemical system of NMC batteries?	We selected mid-sized SUV electric vehicles as this vehicle type represents the most sold EV category across all EV-relevant regions, according to the IEA 2024 report (see Figure below). A mid-sized model was chosen over a large-sized one to yield more balanced results and avoid extremes.

		Breakdown of battery electric car sales in selected countries and regions by segment, 2018-2023 Open ↗ <small>IEA. Licence: CC BY 4.0</small> ● Small car ● Medium car ● Large car ● SUV ● Pick-up truck IEA (2024), Link Furthermore, we opted for high-density NMC (vis-à-vis LFP) cathode chemistries because they remain the predominant choice for these mid-sized models, as per the IEA 2024 findings. The chemical configuration of the traction battery in our study includes NMC811 battery cells. It is important to note that we have integrated these battery cells (the functional unit of Figures 2 and 3) to the battery pack level which is then installed in EVs. For this integration, we use data from ecoinvent, which in turn relies on open-access documents from Dai et al. (2017, 2018) from the Argonne National Laboratory. This step is detailed in the “NMC battery contribution” sheet of the “Breakeven_calculation_BEV_ICE.xlsx” Excel file available in our repository.
3.13	Line 273. What exactly is "the evidence"? Are there any referenced literature or reports? Please list them.	We appreciate the reviewer's request for clarification regarding "the evidence" mentioned in our manuscript. By "the evidence," we were referring to the regional probability density functions (coloured lines) illustrated in Figures 2a and 3a, which show considerable overlap. This overlap demonstrates that the production country of battery cells does not largely determine the cells'

		CF. We have clarified this point in the manuscript by adjusting the sentence in question and being more precise: page 12, line 288 From an environmental policy perspective, the similarities of regional CF profiles (coloured lines in Figure 2/3a) suggest [...]
3.14	The emissions method proposed in the abstract of this paper, which is associated with cost, is barely addressed in the discuss section.	We appreciate the reviewer's observation regarding the coverage of the cost-based methodology in the discussion section and agree that both the contributions of this approach and its limitations warrant more extensive discussion. We have modified the relevant text passages in the discussion section accordingly to provide a more comprehensive analysis: page 14, line 358 While this work proposes a novel cost-based methodology and contributes to the breadth and depth of CF analyses of LIB and their materials, there remains a pressing need for further research. As this methodology does not rely on time-consuming data collection from mining and refining company reports, it can be more easily scaled to additional commodities and panel datasets. Critically, other environmental impacts such as water usage, and biodiversity loss of mining through land use change should be quantified with similar granularity.^{43,51-53} Applying the proposed cost-based methodology to impacts like biodiversity is less straightforward than for global warming and thus warrants future work. [...]
3.15	Line 294-297. Without considering upstream materials, "the CF profiles of Chinese-made LIBs largely coincide with those manufactured in the US or Europe" seems to differ from current research.	We are grateful for the reviewer's correct contextualisation of our findings, particularly in highlighting the discrepancy between the general notion in the field and our novel insights. We attribute this discrepancy to two primary factors: (1) the predominant influence of material choices over production locations, and (2) the reliance on national averages for grid CO2 intensities in most existing comparisons, rather than using more spatially explicit data. Notably, certain regions within China exhibit relatively low grid CO2 intensities, as illustrated in Figure 4b. Without accounting for this, it could lead to incorrect assumptions that all Chinese batteries inherently have a higher CF than those produced in the US or Europe. We have amended the text section to reflect this discussion and incorporate the reviewer's observation more clearly. page 12, line 311 [...] With the caveat of no regional materials preferences of battery producers, the CF profiles of Chinese-made LIBs largely coincide with those manufactured in the US or

		Europe. This new insight can be explained with materials' CF (variance) influence and the spatially-explicit grid CO2 intensities beyond national averages. [...]
3.16	Line 323-324. Please confirm whether the "low-carbon lithium carbonate from..." mentioned is indeed represented in Figure 4b. Verify the entire text and make any necessary revisions.	We thank the reviewer for spotting this mistake. We were, in fact, referring to Figure 4c. Necessary revisions have been made to ensure accuracy throughout the text, although – for the sake of brevity – these changes are not detailed in this response document.
3.17	Line 344-346. Nickel is also not LIB-specific and used in much larger quantities in stainless steels. Why does the author focus on studying it?	This is an important discussion point – we are grateful that this came up during the review. The inclusion of nickel in this study is justified for two main reasons: Firstly, the carbon footprint impact of nickel in lithium-ion batteries (LIBs) is substantial. Therefore, even if the entire nickel supply chain were not specific to LIBs, it would still be crucial to include it due to its environmental significance. This differs from the materials mentioned in line 379f. Secondly, our study focuses exclusively on nickel that could theoretically be processed into high-purity nickel sulfate, which is used in LIBs. The processing routes for nickel chemicals, such as nickel sulfate, differ fundamentally from those used for stainless steel, thereby making it LIB-specific and justifying a detailed analysis. For a deeper understanding of how we delineate the nickel supply chain for this purpose, please refer to SI Section 1.7. To address the reviewer's comment and increase clarity, we added the following sentence to the manuscript: page 15, line 379 The battery material selection is based on CF contribution and LIB-specificity. Copper, aluminium, phosphate, manganese, and iron (sulfate) are not LIB-specific and are used in much larger quantities in other sectors. The production routes for Nickel chemicals used in LIBs differ fundamentally from those in the steel industry, the largest Nickel consumer³¹. Furthermore, manganese, and iron (sulfate) have a negligible CF contribution as shown in Figure 3b. [...]
3.18	Line 452-454. It is suggested that the content of "expert interviews and educated guesses" be supplemented to enhance credibility.	We thank the reviewer for directing our attention to the need for enhanced credibility regarding the use of "expert interviews and educated guesses." Following the reviewer's suggestion, we have clarified the sentence and made the number of estimates explicit (previously "educated guesses"). The revised section now reads: page 19, line 497 [...] For all input parameters, we define uncertainty distributions based on expert interviews, own simulations or multiple data sources. Where only single data points were

		available, we assumed triangular distributions with +-5% (diesel prices) and +-10% (reagent prices). [...]
3.19	Line 567-578. Please provide the data on the provincial- (CN), state- (US), and country-level (Europe, JP, KR) understanding of announced LIB production capacities of NMC and LFP until 2030.	We appreciate the reviewer's request for detailed data on provincial (CN), state (US), and country-level (Europe, JP, KR) LIB production capacities. Unfortunately, we are unable to disclose proprietary data from BloombergNEF (BNEF), which is our primary data source for this. However, the BNEF data aligns closely with Table S3 of Kallitsis et al., who collated open-source battery manufacturing capacity announcements. BNEF uses a consistent methodology across all regions, and some of their data points include specific information on battery chemistry (LFP vs. NMC). For consistency and reliability, we have chosen BNEF as our main data source, corroborating it as necessary, as detailed in the section "Collating Regional Announced Battery Production Capacity" of our manuscript. Additionally, to ensure operability of our method and code for readers without access to the proprietary data bases, our revised repository now features additional open-access data sources for electricity prices for industrial consumers (statista.com and globalpetrolprices.com for BloombergNEF data), diesel prices (iea.org and globalpetrolprices.com for BloombergNEF data) and grid CO2 intensities (ourworldindata.com for ecoinvent). While these sources provide insights similar to the proprietary data, they are less spatially detailed, which is why we rely on proprietary data for our analysis. It is important to note that in our analysis, the absolute capacity numbers do not shape the results as much as the capacity ratios do. Within regions, different capacity ratios of provinces (CN), states (US), and countries (Europe, JP, KR) alongside their local carbon grid intensities determine the shape of the regional probability density functions (coloured lines in Figures 2/3a) and the distribution of the regional jitter plot (coloured dots in Figures 2/3a). Capacity ratios between regions are crucial for the global CF distribution (black line in Figures 2/3a) and the global jitter plot (black dots in Figures 2/3a). Note that, for this analysis, we use tracked battery announcements that do not necessarily reflect actual capacity build-up by 2030.